# Stein Random Feature Regression

**Houston Warren**[1]       **Rafael Oliveira**[2]       **Fabio Ramos**[1,3]

[1]School of Computer Science, The University of Sydney, Sydney, Australia
[2]DATA61, CSIRO, Sydney, Australia,
[3]NVIDIA, USA,

## Abstract

In large-scale regression problems, random Fourier features (RFFs) have significantly enhanced the computational scalability and flexibility of Gaussian processes (GPs) by defining kernels through their spectral density, from which a finite set of Monte Carlo samples can be used to form an approximate low-rank GP. However, the efficacy of RFFs in kernel approximation and Bayesian kernel learning depends on the ability to tractably sample the kernel spectral measure and the quality of the generated samples. We introduce Stein random features (SRF), leveraging Stein variational gradient descent, which can be used to both generate high-quality RFF samples of known spectral densities as well as flexibly and efficiently approximate traditionally non-analytical spectral measure posteriors. SRFs require only the evaluation of log-probability gradients to perform both kernel approximation and Bayesian kernel learning that results in superior performance over traditional approaches. We empirically validate the effectiveness of SRFs by comparing them to baselines on kernel approximation and well-known GP regression problems.

## 1   INTRODUCTION

Gaussian Processes (GPs) are highly regarded in machine learning for their nonparametric regression capabilities. Their sustained prominence, despite the emergence of competitive alternative regression frameworks, stems from their principled approach to modeling uncertainty and the flexibility to incorporate domain-specific inductive biases via kernel covariance functions.

Despite their strengths, the Achilles heel to the GP method is their $\mathcal{O}(N^3)$ computational complexity with respect to the number of data points $N$. To address this, numerous low-rank and sparse methodologies have been developed that seek to preserve GP advantages while mitigating their computational footprint. Notably, random Fourier features (RFFs) [Rahimi and Recht, 2008] and their use in sparse spectrum GPs (SSGP) [Lázaro-Gredilla et al., 2010], represent leading efforts in this domain.

Applying Bochner's theorem [Rudin, 2011], RFF methods model stationary kernels as expectations under a spectral density $\pi(\boldsymbol{\omega})$:

$$k(\mathbf{x}, \mathbf{x}') = \int_{\mathbb{R}^d} \pi(\boldsymbol{\omega}) e^{-i\boldsymbol{\omega}^{\mathsf{T}}(\mathbf{x}-\mathbf{x}')} d\boldsymbol{\omega}, \quad \mathbf{x}, \mathbf{x}' \in \mathbb{R}^d. \quad (1)$$

Given a spectral density $\pi$, $k$ can then be approximated using a finite set of $R \ll N$ Monte Carlo samples $\boldsymbol{\omega} \sim \pi(\boldsymbol{\omega})$, thereby enabling efficient low-rank GP inference.

Approximation of prevalent kernels with known spectral distributions (an example of which being the radial basis function (RBF) and Gaussian spectral measure pair) hinges on the spectral distribution's sampling method. Quasi Monte Carlo (QMC) sampling, noted for its superior approximation accuracy, is effective when $\pi(\boldsymbol{\omega})$ has an accessible inverse-CDF, a condition not met by many common kernels.

RFFs also enable a flexible kernel learning scheme through direct optimization of the $R$ finite samples of $\pi(\boldsymbol{\omega})$ as hyperparameters, offering a pathway to empirically approximate optimal stationary kernels directly from data. However, such schemes are generally susceptible to overfitting [Tan et al., 2016].

A natural remedy is to instead learn a Bayesian posterior over frequencies $p(\boldsymbol{\omega}|D)$ or spectral measure $P(\pi|D)$, but this approach in general does not yield tractable inference. Recent advances have explored MCMC and mean-field variational inference (VI) for approximate kernel posterior inference [Hensman et al., 2018, Miller and Reich, 2022], which respectively offer downsides in computational expense and restricted prior selection on the spectral measure.

Separate to the rise of the RFF and SSGP paradigms has been the growth of particle-based sampling techniques, the

bellwether for which is Stein variational gradient descent (SVGD) [Liu and Wang, 2016], which blends the strengths of MC and VI methodologies. SVGD iteratively refines a set of particles to more closely approximate a target distribution $p$ through gradient descent on the Kullback-Liebler divergence. Crucially, SVGD leverages only gradient evaluations of a target's unnormalized log density. This facilitates efficient sampling from complex Bayesian posteriors previously deemed intractable.

For RFFs, where kernels are approximated through particle samples $\boldsymbol{\omega}$ of a spectral measure $\pi(\boldsymbol{\omega})$, the application of SVGD presents a novel intersection of ideas. Despite their intuitive relationship, the combination of these techniques has received limited attention in literature. In this paper, we make an initial step towards fusing these fields with the presentation of Stein random features (SRFs), which leverages SVGD for fitting, learning, and performing approximate posterior inference on RFF spectral measures and their corresponding kernels. This approach offers novel flexibility and performance advantages, and a significant motivation for this work is to inspire further investigation into the confluence of these methods. We list our contributions as follows:

**Contributions**

- **SVGD Inference for RFFs:** We propose a novel application of Stein variational gradient descent to improve the accuracy of low-rank kernel approximations by utilizing only gradient evaluations of the kernel's spectral measure.

- **Mixture Stein Random Features (M-SRFR):** Extending beyond kernel approximation, we introduce a Bayesian inference framework which uses SVGD to efficiently generate diversified approximate posterior samples of empirical kernel spectral measures.

- **Empirical Benchmarks:** We provide evaluations on common benchmarks in order to demonstrate the flexibility and efficacy of our methods.

## 2 PRELIMINARIES

This section outlines the necessary preliminaries used in the derivation of Stein random features and mixture Stein random feature regression, assuming a baseline familiarity with Gaussian processes (GPs) and kernel covariances. For a thorough review, refer to Rasmussen and Williams [2006].

### 2.1 GAUSSIAN PROCESSES

Gaussian processes (GPs) [Rasmussen and Williams, 2006] are a Bayesian non-parametric regression method that define a distribution over functions. A zero-mean GP prior

$f \sim \mathcal{GP}(0, k_{\boldsymbol{\theta}})$ is uniquely defined by its covariance function $k_{\boldsymbol{\theta}}$, which is specified by its own hyperparameters. Given observations $\mathbf{y} = f(\mathbf{X}) + \boldsymbol{\epsilon}$, at a set of inputs $\mathbf{X} = \{\mathbf{x}_i\}_{i=1}^N \subset \mathbb{R}^d$, assuming $\boldsymbol{\epsilon} \sim \mathcal{N}(0, \sigma^2 \mathbf{I})$, a GP model predicts $\mathbf{f}_* := f(\mathbf{X}_*) \sim \mathcal{N}(\boldsymbol{\mu}_*, \boldsymbol{\Sigma}_*)$ at any $\mathbf{X}_*$ as:

$$\boldsymbol{\mu}_* = \mathbf{K}_{*\mathbf{x}}(\mathbf{K}_{\mathbf{xx}} + \sigma^2 \mathbf{I})^{-1}\mathbf{y}, \tag{2}$$

$$\boldsymbol{\Sigma}_* = \mathbf{K}_{**} - \mathbf{K}_{*\mathbf{x}}(\mathbf{K}_{\mathbf{xx}} + \sigma^2 \mathbf{I})^{-1}\mathbf{K}_{\mathbf{x}*}, \tag{3}$$

where $\mathbf{K}_{\mathbf{xx}} = k_{\boldsymbol{\theta}}(\mathbf{x}, \mathbf{x}'), \forall \mathbf{x}, \mathbf{x}' \in \mathbf{X}$. Kernel and GP hyperparameters $\boldsymbol{\theta}$ are usually estimated by minimising the negative log-marginal likelihood (NLL)[1]:

$$\mathcal{L}(\boldsymbol{\theta}) = \frac{1}{2}\log|\mathbf{K}_{\mathbf{xx}} + \sigma^2 \mathbf{I}| + \frac{1}{2}\mathbf{y}^\top(\mathbf{K}_{\mathbf{xx}} + \sigma^2 \mathbf{I})^{-1}\mathbf{y}$$
$$+ \frac{N}{2}\log(2\pi). \tag{4}$$

The critical limitation of GPs is the $\mathcal{O}(N^3)$ complexity due to Gram matrix $\mathbf{K}$ inversion, which for large $N$ grows computationally intractable.

### 2.2 RANDOM FOURIER FEATURES AND SPARSE SPECTRUM GPS

The computational disadvantages of GPs on large datasets have led to significant interest in low-rank approximations, among which random Fourier features (RFFs) [Rahimi and Recht, 2008] and sparse spectrum Gaussian processes (SS-GPs) [Lázaro-Gredilla et al., 2010] have been significant developments. These approaches derive from Bochner's theorem, which establishes the connection between shift-invariant kernels and non-negative spectral measures. The formulation used here follows the presentation given in Warren et al. [2022]:

**Theorem 1** (Bochner's theorem [Rudin, 2011]). *A shift-invariant kernel $k(\mathbf{x}, \mathbf{x}') = k(\mathbf{x} - \mathbf{x}')$ is positive-definite if and only if it is the Fourier transform of a non-negative measure.*

Bochner's theorem implies that kernels can be uniquely defined through probability measures such that kernel learning can be reframed as learning spectral measures.

**Random Fourier Features:** RFFs propose that we can form finite rank approximations to kernels using Monte Carlo samples of their spectral measure $\pi(\boldsymbol{\omega})$:

$$k(\mathbf{x} - \mathbf{x}') = \int_{\mathbb{R}^d} \pi(\boldsymbol{\omega})e^{i\boldsymbol{\omega}^\top(\mathbf{x}-\mathbf{x}')}\,d\boldsymbol{\omega},$$
$$= \int_{\mathbb{R}^d} \pi(\boldsymbol{\omega})\cos(\boldsymbol{\omega}^\top(\mathbf{x} - \mathbf{x}'))\,d\boldsymbol{\omega} \tag{5}$$
$$\approx \frac{1}{R}\sum_{r=1}^R \cos(\boldsymbol{\omega}_r^\top(\mathbf{x} - \mathbf{x}')),$$

---

[1]In NLL equations, $\pi$ denotes the usual irrational constant arising from the entropy of Gaussian distributions.

An alternative representation is as the dot-product between trigonometric basis functions $k(\mathbf{x} - \mathbf{x}') \approx \Phi(\mathbf{x})^\mathsf{T}\Phi(\mathbf{x}')$:

$$\Phi(\mathbf{x}) = \frac{\sqrt{2}}{\sqrt{2R}}\begin{bmatrix} \cos(\boldsymbol{\omega}_1^\mathsf{T}\mathbf{x}) \\ \sin(\boldsymbol{\omega}_1^\mathsf{T}\mathbf{x}) \\ \vdots \\ \cos(\boldsymbol{\omega}_R^\mathsf{T}\mathbf{x}) \\ \sin(\boldsymbol{\omega}_R^\mathsf{T}\mathbf{x}) \end{bmatrix}. \tag{6}$$

**Sparse Spectrum Gaussian Processes:** SSGPs leverage the RFFs to form a low-rank GP approximation, where the GP predictive equations and log-likelihood are given by:

$$\boldsymbol{\mu}_* = \Phi(\mathbf{X}_*)^\mathsf{T}\mathbf{A}^{-1}\Phi(\mathbf{X})\mathbf{y}, \tag{7}$$

$$\boldsymbol{\Sigma}_* = \sigma^2\Phi(\mathbf{X}_*)^\mathsf{T}\mathbf{A}^{-1}\Phi(\mathbf{X}_*), \tag{8}$$

$$\begin{aligned} \log p(\mathbf{y}|\boldsymbol{\theta}) = &-\frac{1}{2\sigma^2}\left[\mathbf{y}^\mathsf{T}\mathbf{y} - \mathbf{y}^\mathsf{T}\Phi(\mathbf{X})^\mathsf{T}\mathbf{A}^{-1}\Phi(\mathbf{X})\mathbf{y}\right] \\ &-\frac{1}{2}\log|\mathbf{A}| - R\log(R\sigma^2) \\ &-\frac{N}{2}\log(2\pi\sigma^2), \end{aligned} \tag{9}$$

where $\Phi(\mathbf{X}) := [\Phi(\mathbf{x}_1), \ldots, \Phi(\mathbf{x}_N)]$ is defined as in Equation 6 and $\mathbf{A}$ is an $R \times R$ matrix defined by:

$$\mathbf{A} = \Phi(\mathbf{X})\Phi(\mathbf{X})^\mathsf{T} + \sigma^2\mathbf{I}, \tag{10}$$

thus reducing the computational complexity of GP inference to $\mathcal{O}(R^3)$, where $R \ll N$ is the number of Fourier features.

## 2.3 STEIN VARIATIONAL GRADIENT DESCENT

Stein variational gradient descent (SVGD) [Liu and Wang, 2016] represents a means of sampling complex distributions that unifies the strengths of MC methods and variational inference (VI) frameworks via a particle-based, gradient-oriented strategy. VI [Bishop, 2006] seeks to approximate an intractable target distribution $p$ from a family of tractable distributions $q \in \mathcal{Q}$ through minimization of the Kullback-Leibler (KL) divergence between $q$ and $p$:

$$q^* \in \underset{q \in \mathcal{Q}}{\arg\min}\, D_{\mathrm{KL}}(q\|p). \tag{11}$$

VI's success hinges on the choice of approximation family $\mathcal{Q}$, which must offer straightforward inference while being adequately flexible to represent an arbitrarily complex $p$.

SVGD differs from VI in that it instead proposes to apply a sequence of transformations $T_{\mathbf{h}_t}(\boldsymbol{\omega}) = \boldsymbol{\omega} + \mathbf{h}_t(\boldsymbol{\omega})$ to particles $\boldsymbol{\omega}$ sampled from an initial distribution $\boldsymbol{\omega} \sim q_0$, steering them towards $p$ by following gradient flows of $D_{\mathrm{KL}}(q\|p)$ [Liu et al., 2019]. Crucially, the variational approximation $q$ is non-parametric, and therefore not confined to a specific family $\mathcal{Q}$. Assuming $\mathbf{h}_t$ is a member of a reproducing kernel Hilbert space $\mathcal{H}_\kappa^d$, the optimal transformation $T_{\mathbf{h}_t}$ has an

analytic expression, resulting in a tractable algorithm for variational inference:

$$\boldsymbol{\omega}_i^{t+1} = \boldsymbol{\omega}_i^t + \epsilon\mathbf{h}_t(\boldsymbol{\omega}_i^t), \tag{12}$$

where $\epsilon$ is a small step-size parameter and $\mathbf{h}_t$ is given by:

$$\mathbf{h}_t(\boldsymbol{\omega}) = \frac{1}{R}\sum_{j=1}^R \kappa(\boldsymbol{\omega}, \boldsymbol{\omega}_j^t)\nabla_{\boldsymbol{\omega}_j^t}\log p(\boldsymbol{\omega}_j^t) + \nabla_{\boldsymbol{\omega}_j^t}\kappa(\boldsymbol{\omega}, \boldsymbol{\omega}_j^t), \tag{13}$$

in which $\kappa : \mathbb{R}^d \times \mathbb{R}^d \to \mathbb{R}$ is a positive-definite kernel function, and $\boldsymbol{\omega}_0^i \sim q_0$, for a given base distribution $q_0$. The first term in Equation 13 serves as an attractive force for particles $\boldsymbol{\omega}_i$ to converge on high density regions of $p$ while the second term serves as a repulsive force that encourages diversity between particles and avoids mode collapse.

SVGD's advantage over VI is that it relies only on the specification of a kernel function $\kappa$ and gradient evaluations of a target's (unnormalized) log probability $\log p(\boldsymbol{\omega})$. Such advantages make SVGD applicable in the approximation of posterior distributions for which it could be challenging to find a suitable variational family of parametric distributions.

## 2.4 FUNCTIONAL KERNEL LEARNING

In the GP framework, optimization of kernel hyperparameters $\boldsymbol{\theta}$ focuses on the GP log-likelihood in Equation 4, aiming to find:

$$\begin{aligned} \boldsymbol{\theta}^* &\in \underset{\boldsymbol{\theta}}{\arg\min}\, -\log p(\mathbf{y}|\boldsymbol{\theta}) \\ &= \underset{\boldsymbol{\theta}}{\arg\min}\, \mathcal{L}(\boldsymbol{\theta}) \end{aligned} \tag{14}$$

By adopting Bochner's theorem, which allows for expressing kernels via their spectrum, one may shift towards optimizing the kernel directly via its spectral measure $\pi(\boldsymbol{\omega})$. This transforms GP optimization into a *functional* objective over the space $\mathcal{P}(\mathbb{R}^d)$ of probability measures on $\mathbb{R}^d$:

$$\pi^* \in \underset{\pi \in \mathcal{P}(\mathbb{R}^d)}{\arg\min}\, \mathcal{L}[\pi] \tag{15}$$

For a finite sample $\boldsymbol{\Omega}_0 := \{\boldsymbol{\omega}_{i,0}\}_{i=1}^R$, where $\boldsymbol{\omega}_{i,0} \sim \pi_0$, one approach is to follow the gradient of the GP NLL $\mathcal{L}[\hat{\pi}_t] = \mathcal{L}(\boldsymbol{\Omega}_t) = -\log p(\mathbf{y}|\boldsymbol{\Omega}_t)$, updating:

$$\boldsymbol{\Omega}_{t+1} = \boldsymbol{\Omega}_t - \epsilon\nabla_{\boldsymbol{\Omega}_t}\mathcal{L}(\boldsymbol{\Omega}_t). \tag{16}$$

Note that $\boldsymbol{\Omega}$ can be simply seen as a matrix, so that $\nabla_{\boldsymbol{\Omega}}\mathcal{L}(\boldsymbol{\Omega})$ is also a matrix. This approach was proposed in the original sparse spectrum Gaussian processes work by Lázaro-Gredilla et al. [2010], which also included the other GP hyperparameters into the same optimization loop. As a result, one obtains a maximum likelihood estimate (MLE) of the kernel and its empirical spectral measure $\hat{\pi} := \frac{1}{R}\sum_{i=1}^R \delta_{\boldsymbol{\omega}_i}$, where $\delta_{\boldsymbol{\omega}}$ represents the Dirac measure at $\boldsymbol{\omega} \in \mathbb{R}^d$.

# 3 RELATED WORK

**Kernel Approximation with RFFs** There has been significant focus on improving the quality of the RFF kernel approximation presented in Equation 5. One avenue considers enhancing the quality MC and QMC samples $\omega_i$ through post-hoc adjustments to reduce variance and bolster approximation quality [Le et al., 2013, Yu et al., 2016, Chang et al., 2017]. Additionally, alternative quadrature techniques, including numerical and Bayesian quadrature, have been proposed as alternative means for the integral approximation of Equation 5 [O'Hagan, 1991, Mutny and Krause, 2018]. A thorough review is available in Liu et al. [2021]. We posit that our SVGD-based approach for kernel approximation, detailed in Section 5.1, offers distinct benefits. It simplifies implementation across spectral measures using gradient evaluations and is underpinned by SVGD's robust theory.

**Spectral Kernel Learning** Bochner's Theorem 1 has motivated a plethora of techniques that conceptualize kernel learning as probabilistic inference on spectral measures. Beyond RFFs, spectral mixture kernels (SMKs) [Wilson and Adams, 2013] represent kernel spectral measures as Gaussian mixture models. Recent advances have generalized and extended the SMK approach to introduce nonstationarity, scalability, and variational inference [Samo and Roberts, 2015, Remes et al., 2017, Shen et al., 2019, Jung et al., 2022].

Concurrently, RFFs have evolved through integration with advanced kernel learning and GP architectures, including deep kernels [Xie et al., 2019, Xue et al., 2019, Mallick et al., 2021], generative adversarial networks [Li et al., 2019], and deep Gaussian process [Cutajar et al., 2017]. To avoid overfitting, Bayesian inference over frequencies $\omega$ using MCMC Miller and Reich [2022] and variational inference [Hensman et al., 2018, Zhen et al., 2020, Cheema and Rasmussen, 2023] has also been proposed.

The methodology we propose compliments many of these advances. M-SRFR can be viewed as simply adding a mixture dimension $M$ to a point estimate of kernel spectral measure parameters $\omega$, with access to gradients of the score function – which nearly all of the aforementioned architectures calculate in training – being the only requirement for implementation.

**Statistical Inference over Functions** Statistical inference over functions, within kernel learning and broader contexts, is an area of active research. Others have considered performing functional inference on kernels like we do here, but differ in that they assign distributional families to the spectral measure priors as GPs [Benton et al., 2019] or Gaussian mixtures [Hamid et al., 2022].

Functional inference is in general a popular research topic

within the context of Bayesian inference and Bayesian neural networks [Wang and Liu, 2019, Sun et al., 2019, Ma and Hernández-Lobato, 2021, D'Angelo et al., 2021, Pielok et al., 2023]. A notable commonality between these approaches is their use of SVGD or other particle-based techniques for inference. These works help to inspire the method we now present, which is a more focused application of the functional inference problem to kernel learning in GPs.

# 4 STEIN RANDOM FEATURES

We now derive our proposed methodologies for operationalizing the theoretical and intuitive connections between RFFs and SVGD. We specifically propose two promising routes:

1. Using SVGD as a sampling mechanism for forming RFF approximations to kernels with known spectral measures.

2. Extending the results of Section 2.4 to propose mixture Stein random features (M-SRFR) for posterior inference over kernel spectral measures.

The former item we leave to Section 5.1, as it represents a straightforward, though nonetheless novel, application of SVGD sampling routines for generating RFF kernel frequencies. Instead, we focus here on motivating the use of SVGD in kernel learning from a theoretical perspective, and subsequently extending to posterior inference.

## 4.1 RECOVERING SVGD THROUGH FUNCTIONAL KERNEL LEARNING

Mallick et al. [2021], deriving a result for deep probabilistic kernel learning applicable in the RFF context, show that with kernel matrices defined in the form of Equation 1, the optimal $\pi^*$ for the functional objective of Equation 15 can be approximated as a non-parametric particle approximation $q^*$ with gradients:

$$\begin{aligned} \nabla_\pi \mathcal{L}[\pi] &\approx \nabla_q \mathcal{L}[q] \\ &\approx \sum_{r=1}^{R} \kappa(\omega_r, \cdot) \nabla_{\omega_r} \mathcal{L}(\omega), \end{aligned} \tag{17}$$

We can observe that the gradient step defined in (17) equates to a particle update similar to the SVGD update in (13), but without a particle repulsion term $\nabla_{\omega_j} \kappa(\omega, \omega_j)$. To directly recover a variant of SVGD through incorporation of the repulsive term, we can use the results of Liu et al. [2019] who show that particle repulsion can be derived from the addition of an entropy regularization term $H[q]$ to the functional marginal log-likelihood:

$$\pi^* \approx q^* = \arg\min_q \mathcal{L}[q] - H[q]. \tag{18}$$

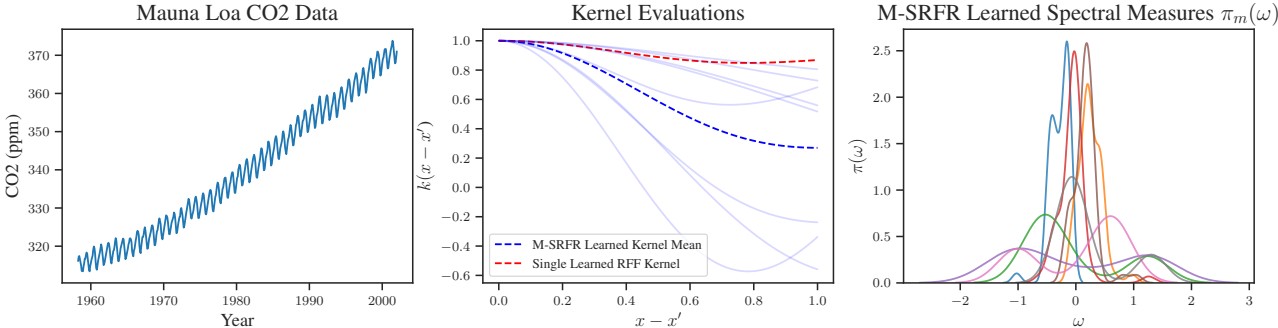

Figure 1: Comparison of traditional RFF Kernel Learning to an M-SRFR posterior with $M = 8$ components

The result of Equation 18 a kernel learning scheme with near equivalence to SVGD, thereby signifying a theoretical convergence between kernel learning on RFFs and SVGD that matches their intuitive connections. Detailed derivations of these properties are provided in the supplement (see Appendix A).

## 4.2 POSTERIORS OVER SPECTRAL MEASURES

Equation 18 represents entropy-regularized maximum-likelihood inference over a kernel spectral measure $\pi(\boldsymbol{\omega})$, which results in a single estimate. We propose to extend these results to instead perform Bayesian inference over spectral measures, leading to a posterior over kernels.

We begin by formulating the posterior over kernel spectral densities $\pi(\boldsymbol{\omega})$. The GP likelihood of observing data $D$ under a kernel $k$ characterized by spectral measure $\pi$ is denoted as $p(D|\pi)$. Keeping aside measure-theoretic formalities and regularity conditions for now, assume we have a prior $P$ over the space of probability distributions $\mathcal{P}(\mathbb{R}^d)$. We can then formulate a posterior over a kernel's spectral measure $\pi$ as:

$$P(\pi|D) \propto p(D|\pi)P(\pi). \tag{19}$$

Taking a similar functional kernel learning approach, Benton et al. [2019] choose to represent $P(\pi)$ by placing a GP prior on $\log \pi(\boldsymbol{\omega})$ and applying Markov chain Monte Carlo for inference over the latent process [Murray and Adams, 2010]. We however adopt a particle-based VI approach via SVGD.

We introduce a variational approximation $Q(\pi) \approx P(\pi|D)$, which will be characterized by an empirical particle distribution. The variational objective is to minimize the KL divergence between the approximate and the true posterior:

$$Q^* \in \arg\min_{Q} D_{\mathrm{KL}}(Q(\pi)||P(\pi|D)). \tag{20}$$

Given initial particles $\pi_m \sim Q_0(\pi)$, our goal is to identify a transformation $T$ which through minor adjustments $\epsilon$ guides particles towards minimizing the KL divergence. Wang and

Liu [2019] demonstrate that, even if $p(D|\pi)$ is a nonlinear functional of $\pi$, the optimal transformation of particles $\pi_m$ can still be given by a Stein update rule similar to (13).

The GP likelihood in (19) is a nonlinear functional of the spectral measure $\pi(\boldsymbol{\omega})$ via the kernel (1), which appears in nonlinear relations in (4). We can thus apply SVGD to iteratively update particles $\pi_m$ to minimize (20) and form an empirical posterior approximation $Q^*$. The difference in this setting versus traditional SVGD is that we view individual particles as probability measures $\pi_m$ themselves, rather than samples of a probability measure over individual points.

## 4.3 INFERENCE IN THE SPACE OF MEASURES

Now we describe a more formal treatment to the inference problem we have at hand. To precisely define a prior over the space of probability measures $\mathcal{P}(\mathbb{R}^d)$, we consider transport maps and their pushforwards, instead of the individual measures directly. Similar to traditional SVGD, let $T_{\mathbf{h}}(\boldsymbol{\omega}) := \boldsymbol{\omega} + \mathbf{h}(\boldsymbol{\omega})$, but now assume that $\mathbf{h}$ follows a stochastic process $\mathcal{SP}$, which defines a prior $P_{\mathcal{F}}$ over the space of functions $\mathcal{F}(\mathbb{R}^d)$ mapping $\mathbb{R}^d$ to itself. For instance, we can have a vector-valued Gaussian process as a prior $\mathbf{h} \sim \mathcal{GP}(\mathbf{0}, \boldsymbol{\Sigma})$, defined by a matrix-valued kernel, e.g., $\boldsymbol{\Sigma}(\boldsymbol{\omega}, \boldsymbol{\omega}') := \kappa_{\mathbf{h}}(\boldsymbol{\omega}, \boldsymbol{\omega}')\mathbf{I}$ [Alvarez et al., 2012]. Given a base measure $\pi_0$, each realisation of the pushforward[2] $\pi_{\mathbf{h}} := T_{\mathbf{h}}\#\pi_0$ defines a probability measure in $\mathcal{P}(\mathbb{R}^d)$. Therefore, the stochastic process $\mathbf{h} \sim \mathcal{SP}$ defines a prior over $\mathcal{P}(\mathbb{R}^d)$ via the corresponding transport maps $T_{\mathbf{h}}$.

Now we formulate an SVGD perspective over the space of vector-valued functions $\mathcal{F} := \mathcal{F}(\mathbb{R}^d)$. Given $\mathbf{g} : \mathbb{R}^d \to \mathbb{R}^d$, let $T_{\mathbf{g}} : \mathcal{F} \to \mathcal{F}$ define a transform on $\mathcal{F}$ such that $T_{\mathbf{g}}(\mathbf{h})(\boldsymbol{\omega}) = \mathbf{h}(\boldsymbol{\omega}) + \mathbf{g}(\mathbf{h}(\boldsymbol{\omega}))$, for all $\boldsymbol{\omega} \in \mathbb{R}^d$. Given a base measure $Q_0$ over $\mathcal{F}$, associated with a stochastic process $\mathbf{h}_0 \sim Q_0$, we aim to apply a sequence of transformations $T_{\mathbf{g}_t}$ to $Q_0$, so that the pushforward $Q_t := T_{\mathbf{g}_t}\#Q_{t-1}$

---

[2]The pushforward of a measure $P$ on $\mathcal{X}$ by a measurable map $T : \mathcal{X} \to \mathcal{Y}$ is defined as the measure $Q := T\#P$ on $\mathcal{Y}$ such that $Q(\mathcal{A}) = P(\{x \in \mathcal{X} \mid T(x) \in \mathcal{A}\})$ for any measurable $\mathcal{A} \subset \mathcal{Y}$.

converges to a target measure $P_*$ on $\mathcal{F}$ as $t \to \infty$. To do so, we follow the gradient flow of the KL divergence:

$$D_{\mathrm{KL}}(Q_t||P_*) = \mathbb{E}_{\mathbf{h} \sim Q_t}\left[\log \frac{\mathrm{d}Q_t}{\mathrm{d}P_*}(\mathbf{h})\right], \quad (21)$$

where $\frac{\mathrm{d}Q_t}{\mathrm{d}P_*}$ is the Radon-Nikodym derivative of $Q_t$ with respect to $P_*$ [Bauer, 1981]. Assuming $Q$ is absolutely continuous w.r.t. $P_*$, the derivative $\frac{\mathrm{d}Q}{\mathrm{d}P_*}$ is well defined. As shown in previous works in the literature of function-space VI [Ma and Hernández-Lobato, 2021], the KL divergence between two stochastic processes is equivalent to:

$$D_{\mathrm{KL}}(Q_t||P_*) = \sup_{n, \mathbf{\Omega}_n} D_{\mathrm{KL}}(q_t(\mathbf{H}_n)||p_*(\mathbf{H}_n)), \quad (22)$$

where $\mathbf{H}_n := \mathbf{h}(\mathbf{\Omega}_n) = [\mathbf{h}(\boldsymbol{\omega}_1), \dots, \mathbf{h}(\boldsymbol{\omega}_n)]^\top$, and $\mathbf{\Omega}_n$ is an $n$-element subset of $\mathbb{R}^d$, or equivalently an $n$-by-$d$ matrix, and $q_t$ and $p_*$ are the joint probability measures on $\mathbb{R}^{n \times d}$ associated with the stochastic processes defined by $Q_t$ and $P_*$, respectively. Given the above, we can now state our theoretical result, which we prove in Appendix B.

**Theorem 2.** *Let $T_{\mathbf{g}}$ be as above, where $\mathbf{g} : \mathbb{R}^d \to \mathbb{R}^d$ is an element of the vector-valued reproducing kernel Hilbert space $\mathcal{H}_\kappa^d$ associated with a positive-definite kernel $\kappa$. Given a probability measure $P_*$ on $\mathcal{F}$, the direction of steepest descent in the KL divergence $D_{\mathrm{KL}}(Q_{\mathbf{g}}||P_*)$ is given by:*

$$\nabla_{\mathbf{g}} D_{\mathrm{KL}}(Q_{\mathbf{g}}||P_*)\big|_{\mathbf{g}=0} =$$
$$- \mathbb{E}_{\mathbf{h} \sim Q}[\kappa(\cdot, \mathbf{H}^*)\nabla_{\mathbf{H}^*} \log p_*(\mathbf{H}^*) + \nabla_{\mathbf{H}^*}\kappa(\cdot, \mathbf{H}^*)], \quad (23)$$

*where $\mathbf{H}^* := \mathbf{h}(\mathbf{\Omega}_{n^*}^*)$, assuming the supremum is reached at $n^*$ and $\mathbf{\Omega}_{n^*}^*$ in Equation 22. In particular, for an empirical base measure $\hat{\pi} := \frac{1}{R}\sum_{i=1}^R \delta_{\boldsymbol{\omega}_i}$ supported on $\mathbf{\Omega}_R = \{\boldsymbol{\omega}_i\}_{i=1}^R$, we have that:*

$$\nabla_{\mathbf{g}} D_{\mathrm{KL}}(Q_{\mathbf{g}}||P_*)\big|_{\mathbf{g}=0} =$$
$$- \mathbb{E}_{\mathbf{H} \sim q(\mathbf{H})}[\kappa(\cdot, \mathbf{H})\nabla_{\mathbf{H}} \log p_*(\mathbf{H}) + \nabla_{\mathbf{H}}\kappa(\cdot, \mathbf{H})], \quad (24)$$

*where $\mathbf{H} := \mathbf{h}(\mathbf{\Omega}_R) \in \mathbb{R}^{R \times d}$.*

This result allows us to apply SVGD steps in the space of measures $\mathcal{P}(\mathbb{R}^d)$ by following the gradient flow of the KL divergence between stochastic processes. We can then identify $P_*$ with $P(\pi|D)$ in the previous section to learn an approximation to the posterior distribution over spectral measures. Moreover, this result also shows us that we can treat inference in the space of measures as inference over matrices, when restricted to empirical measures. Further discussion about the theoretical result in Theorem 2 and its application to our problem formulation is deferred to Section B.3 in the appendix.

---

**Algorithm 1** Mixture Stein Random Feature Regression (M-SRFR)

**Require:** Dataset $D$, GP kernel $k$ parametrized by $M$ RFF frequency matrices $\{\mathbf{\Omega}_m\}_{m=1}^M$, SVGD kernel $\kappa$, particle prior $p(\mathbf{\Omega})$, step size $\epsilon$, hyperparamter $\alpha$, and number of iterations $T$.

1: **for** t = 1 to T **do**
2:     Compute gradient $\nabla_{\mathbf{\Omega}_m} \log p(D|\mathbf{\Omega}_m)p(\mathbf{\Omega}_m)$ with $p(D|\mathbf{\Omega}_m)$ from (9) for all $m \in \{1, \dots, M\}$.
3:     **for** each $\mathbf{\Omega}_m, m \in \{1, \dots, M\}$ **do**
4:         **for** each $\mathbf{\Omega}_j, j \in \{1, \dots, M\}$ **do**
5:             Compute the kernel value $\kappa(\mathbf{\Omega}_m, \mathbf{\Omega}_j)$ from (66) and gradient $\nabla_{\mathbf{\Omega}_j}\kappa(\mathbf{\Omega}_m, \mathbf{\Omega}_j)$ from (67)
6:         **end for**
7:         Apply M-SRFR update rule $\mathbf{\Omega}_m^{t+1} \leftarrow \mathbf{\Omega}_m^t$ according to 26
8:     **end for**
9: **end for**
10: **Output:** Learned kernel $k$ with frequencies $\{\mathbf{\Omega}_m^T\}_{m=1}^M$

---

## 4.4 MIXTURE STEIN RANDOM FEATURE REGRESSION

As above, let $\mathbf{\Omega}_m$ represent a set of samples $\{\boldsymbol{\omega}_{i,m}\}_{i=1}^R$ drawn from a spectral measure $\pi_m$. In the finite-particle setting, we can associate the spectral measure posterior in (19) with a posterior over the empirical representation of $\pi$ by a frequency matrix $\mathbf{\Omega}$:

$$\begin{aligned} P(\pi|D) &\approx P(\mathbf{\Omega}|D), \\ p(\mathbf{\Omega}|D) &\propto p(D|\mathbf{\Omega})p(\mathbf{\Omega}). \end{aligned} \quad (25)$$

Now we have a prior over matrices $p(\mathbf{\Omega})$, which can be constructed by applying priors over the individual row vectors in it, each representing a spectral frequency. For example, standard Gaussians and mixtures of them can be trivially extended to the matrix-variate setting. In any case, the methodology we derive is agnostic to the choice of prior, as long as it is differentiable with respect to $\mathbf{\Omega}$, making it flexible to incorporate a variety of prior knowledge or smoothness assumptions on the spectral distribution.

With the theoretical underpinnings of Theorem 2, we can substitute $\pi$ with $\mathbf{\Omega}$ into the Stein update rules in Equation 13. This forms the key component of our proposed method, *Mixture Stein Random Feature Regression* (M-SRFR).

**Definition 3** (Mixture Stein Random Feature Regression)**.** Given an initial set of $M$ frequency matrices $\mathbf{\Omega}_m \in \mathbb{R}^{R \times d}$, where each $\boldsymbol{\omega}_{i,m} \sim q_0(\boldsymbol{\omega})$, M-SRFR defines an update:

$$\mathbf{\Omega}_m^{t+1} = \mathbf{\Omega}_m^t + \frac{\epsilon}{M}\sum_{j=1}^M [\kappa(\mathbf{\Omega}_m, \mathbf{\Omega}_j)\nabla_{\mathbf{\Omega}_j} \log p(\mathbf{\Omega}_j)$$
$$+ \alpha \nabla_{\mathbf{\Omega}_j}\kappa(\mathbf{\Omega}_m, \mathbf{\Omega}_j)], \quad (26)$$

where $\alpha$ is a temperature parameter, and the score gradient $\nabla_{\boldsymbol{\Omega}_m} \log p(\boldsymbol{\Omega}_m)$ can be separated into:

$$\nabla_{\boldsymbol{\Omega}_m} \log p(D|\boldsymbol{\Omega}_m) + \sum_{i=1}^{R} \nabla_{\boldsymbol{\omega}_{i,m}} \log p(\boldsymbol{\omega}_{i,m}), \quad (27)$$

which represents the gradient of the SSGP likelihood (9) given the RFF frequency matrix $\boldsymbol{\Omega}_m$ and the sum of a frequency prior $p(\boldsymbol{\omega})$ over the component frequencies $\{\boldsymbol{\omega}_{i,m}\}_{i=1}^{R} = \boldsymbol{\Omega}_m$. The inter-particle kernel is given by the kernel between the rows of each matrix of frequencies:

$$\kappa(\boldsymbol{\Omega}, \boldsymbol{\Omega}') = \begin{bmatrix} \kappa(\boldsymbol{\omega}_1, \boldsymbol{\omega}_1') & \dots & \kappa(\boldsymbol{\omega}_1, \boldsymbol{\omega}_R') \\ \vdots & \ddots & \vdots \\ \kappa(\boldsymbol{\omega}_R, \boldsymbol{\omega}_1') & \dots & \kappa(\boldsymbol{\omega}_R, \boldsymbol{\omega}_R') \end{bmatrix}. \quad (28)$$

M-SRFR is summarized in Algorithm 1, and we provide further implementation and gradient calculation details in Appendix B.4. The construction of $\boldsymbol{\Omega}_m$ via frequencies $\{\boldsymbol{\omega}_{i,m}\}_{i=1}^{R}$ implies that gradient updates to $\boldsymbol{\Omega}_m$ directly modify its constituent frequencies. Collectively, the parameters of the M-SRFR model form a tensor with dimensions $M \times R \times d$. An example comparison of M-SRFR and traditional RFF learning is presented in Figure 1.

M-SRFR performs approximate Bayesian posterior inference over $P(\pi)$ when the temperature parameter $\alpha = 1$. However, when working with a small number of mixture components $M$, we observed empirical benefit to including $\alpha$ in the hyperparameter training routine to regulate the strength of the repulsive force.

For making predictions with M-SRFR, summarized in Algorithm 2, we combine the $M$ individual mixture predictive means and covariances into a single predictive distribution. We do so using properties of Gaussian mixtures, which allows for calculation of an overall mean and covariance as a uniformly-weighted aggregate of the mixture components.

### 4.5 COMPLEXITY AND EXTENSIBILITY

Conceptually, M-SRFR orchestrates an ensemble of $M$ SS-GPs, promoting diversity through kernel repulsion term $\nabla_{\boldsymbol{\Omega}_j} \kappa(\boldsymbol{\Omega}_m, \boldsymbol{\Omega}_j)$ and preventing mode collapse. The sparsity of SSGPs ensures computational feasibility, with an increase in complexity from $\mathcal{O}(R^3)$ to $\mathcal{O}(MR^3)$, and in practice we find that performance increases are present even when using a small number of $M \ll R$ mixture components. Additionally, empirical evidence in Section 5 suggests that the mixture approach of M-SRFR outperforms RFFs with an equivalent complexity using $R^* = \sqrt[3]{MR^3}$ features.

M-SRFR's versatility extends to a broad spectrum of kernel learning applications, including nonstationary Fourier features [Ton et al., 2018], spectral kernel learning [Wilson and Adams, 2013], and GPs with non-Gaussian likelihoods.

---

**Algorithm 2** M-SRFR Prediction on New Inputs

**Require:** Dataset $D$, new inputs $\mathbf{X}^*$, and trained M-SRFR GP kernel $k$ parametrized by $M$ RFF frequency matrices $\{\boldsymbol{\Omega}_m\}_{m=1}^{M}$.
1: **for** each frequency matrix $\boldsymbol{\Omega}_m, m \in \{1, \dots, M\}$ **do**
2:     Given inputs $\mathbf{X}^*$, compute SSGP prediction mean $\boldsymbol{\mu}_m^*$ from (7) and covariance $\boldsymbol{\Sigma}_m^*$ from (8) using an RFF kernel defined by frequencies $\boldsymbol{\Omega}_m$
3: **end for**
4: Calculate $\boldsymbol{\mu}^*$ and $\boldsymbol{\Sigma}^*$ with:

$$\boldsymbol{\mu}^* = \frac{1}{M} \sum_{m=1}^{M} \boldsymbol{\mu}_m^*$$

$$\boldsymbol{\Sigma}^* = \frac{1}{M} \sum_{m=1}^{M} \boldsymbol{\Sigma}_m^* + (\boldsymbol{\mu}_m^* - \boldsymbol{\mu}^*)(\boldsymbol{\mu}_m^* - \boldsymbol{\mu}^*)^T$$

5: **Output:** predictions $\mathbf{y}^* \sim \mathcal{N}(\boldsymbol{\mu}^*, \boldsymbol{\Sigma}^* | \mathbf{X}^*, D)$

---

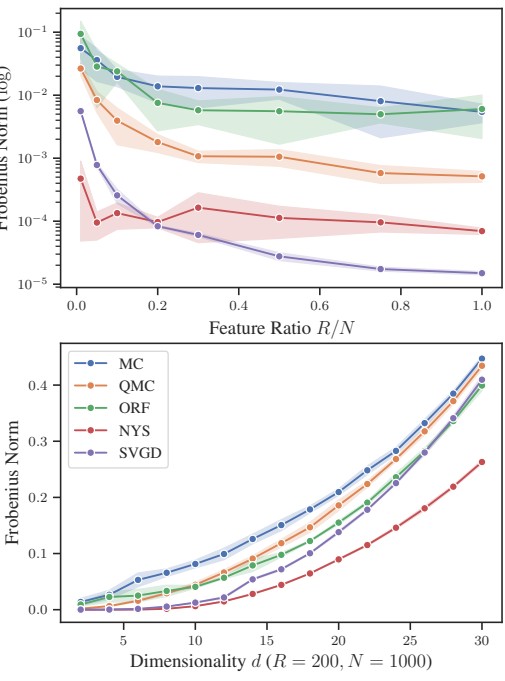

Figure 2: Kernel approximation error and standard deviations over 10 random seeds.

This adaptability stems from SVGD's sole reliance on gradient evaluations of the score function, readily facilitated by widely used auto-differentiation tools.

## 5 EXPERIMENTS

We now demonstrate the efficacy of both the SVGD approach to approximating kernels with known spectral measures as well as the performance of our M-SRFR method on

Table 1: UCI Regression Benchmarks RMSE and NLPD with standard deviations over 10 random seeds.

| | airfoil $R = 100, M = 6$ | concrete $R = 100, M = 6$ | energy $R = 50, M = 10$ | wine $R = 100, M = 10$ |
|---|---|---|---|---|
| | | *RMSE* | | |
| SVGP | $2.36 \pm 0.24$ | $6.35 \pm 0.69$ | $2.72 \pm 0.17$ | $0.62 \pm 0.04$ |
| SSGP-RBF | $2.90 \pm 0.60$ | $5.74 \pm 0.58$ | $0.48 \pm 0.03$ | $0.82 \pm 0.08$ |
| SSGP | $2.41 \pm 0.53$ | $5.03 \pm 0.74$ | $0.37 \pm 0.05$ | $0.87 \pm 0.04$ |
| SSGP-$R^*$ | $2.54 \pm 1.09$ | $4.88 \pm 0.65$ | $0.36 \pm 0.03$ | $0.69 \pm 0.06$ |
| SSGP-SVGD | $2.50 \pm 0.59$ | $5.51 \pm 0.54$ | $0.40 \pm 0.09$ | $0.76 \pm 0.06$ |
| M-SRFR (Ours) | $\mathbf{1.88 \pm 0.27}$ | $\mathbf{4.13 \pm 0.72}$ | $\mathbf{0.29 \pm 0.04}$ | $\mathbf{0.59 \pm 0.04}$ |
| | | *Negative Log Predictive Density (NLPD)* | | |
| SVGP | $487.0 \pm 203.4$ | $272.8 \pm 120.3$ | $993.6 \pm 205.3$ | $2334.9 \pm 457.0$ |
| SSGP-RBF | $780.5 \pm 457.0$ | $36.3 \pm 11.8$ | $-249.4 \pm 8.7$ | $791.1 \pm 160.7$ |
| SSGP | $216.0 \pm 159.9$ | $23.1 \pm 14.0$ | $-288.0 \pm 23.9$ | $783.9 \pm 82.3$ |
| SSGP-$R^*$ | $213.4 \pm 369.6$ | $20.2 \pm 12.0$ | $-293.9 \pm 17.0$ | $404.0 \pm 74.3$ |
| SSGP-SVGD | $4166.6 \pm 2885.5$ | $29.4 \pm 10.0$ | $-261.7 \pm 58.8$ | $16924.9 \pm 2615.7$ |
| M-SRFR (Ours) | $454.3 \pm 134.5$ | $113.9 \pm 77.3$ | $-283.7 \pm 38.4$ | $1882.5 \pm 205.3$ |

common GP regression UCI benchmarks [Dua and Graff, 2017]. Code has been made available[3].

## 5.1 STEIN RANDOM FEATURES FOR KERNEL APPROXIMATION

A notable yet less-emphasized contribution we introduce is leveraging SVGD-generated samples as frequencies $\omega$ in RFFs for accurately reconstructing kernels with known spectral distributions. While QMC [Morokoff and Caflisch, 1995] sampling typically yields high-quality reconstructions by necessitating tractable inverse-CDFs of kernel spectral measures – a requirement not met by many common kernels – SVGD circumvents this by merely requiring the spectral measure's score gradient.

In a comparative experiment focused on the Gaussian (RBF) kernel, we evaluate the approximation quality of randomized kernel Gram matrices $\mathbf{K}$ using RFFs with varied sampling techniques as well as other common low-rank kernel approximation methods. Specifically, we benchmark Matrix-SVGD [Wang et al., 2019] sampling against MC, QMC, orthogonal random features (ORF) [Yu et al., 2016], and Nyström approximation (NYS) [Yang et al., 2012]. Results are demonstrated in Figure 2, where we use as a metric the Frobenius norm $\frac{||\mathbf{K} - \hat{\mathbf{K}}||}{||\mathbf{K}||}$ between Gram approximation $\hat{\mathbf{K}}$ and true Gram matrix $\mathbf{K}$.

The results underscore SVGD's strong approximation capabilities over other sampling techniques across different ranks $R$, and notably, SVGD outperforms data-dependent methods like Nystroem as $R$ increases. Across data dimensionality $d$, SVGD scales better than existing sampling

based approaches, though not as efficiently as Nyström. Nonetheless, given the challenges kernel methods face in high-dimensional spaces, practitioners likely resort to dimensionality reduction before applying kernel techniques in such high-dimensional settings.

## 5.2 UCI REGRESSION BENCHMARKS

We evaluate M-SRFR on a variety of regression problems from the UCI data repository [Dua and Graff, 2017], with baselines of sparse variational Gaussian processes [Hensman et al., 2015] (SVGP), SSGPs with an RFF Gaussian kernel (SSGP-RBF), an SSGP with frequencies as hyperparameters, and an SSGP trained using the entropy-regularized functional kernel learning approach described in Section 2.4 (SSGP-SVGD), for which more details can be found in Wang and Liu [2019]. Additionally we introduce SSGP-$R^*$, which has $R^* = \sqrt[3]{MR^3}$ frequency samples, matching M-SRFR's computational complexity for $M$ mixture components.

All models and baselines are given a proper hyperparameter optimization treatment. The results in Table 1, where $M$ represents the number of mixture components used in the M-SRFR model, highlight M-SRFR's superior RMSE performance, particularly when contrasted with SSGP-$R^*$, underscoring the mixture approach's advantage over simply enhancing RFF feature count. We employed wide Gaussian priors on the M-SRFR frequencies, and posit that specialized priors tailored to data characteristics may further enhance performance.

**Limitations in Uncertainty Calibration** NLPD, which measures uncertainty calibration, results vary. All models exhibit high variance across seeds, hence we do not bold

---

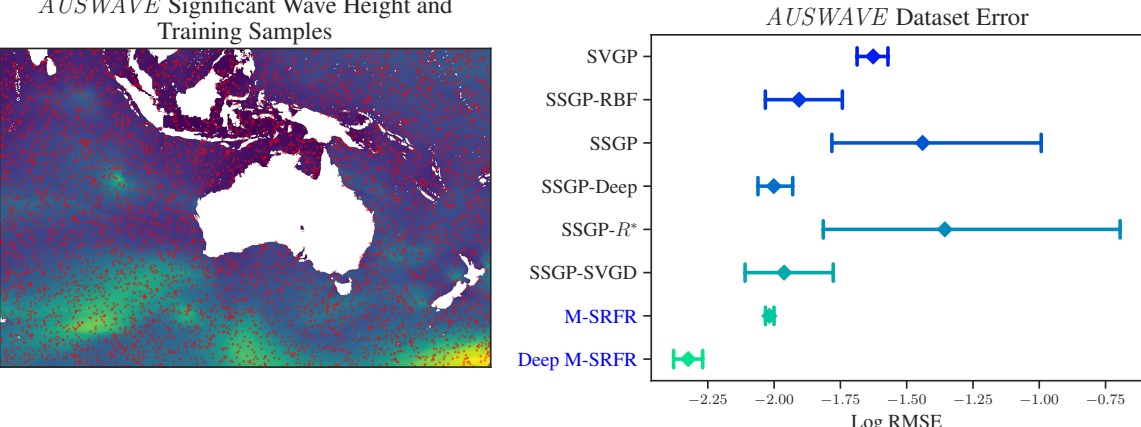

Figure 3: *AUSWAVE* dataset and error with contributed methods labeled in blue.

results. M-SRFR is competitive in NLPD on many datasets but is outperformed by simpler baselines that tend to make wider and less mean-accurate predictions, as we demonstrate in Appendix D. The drop-off in performance from RMSE to NLPD for M-SRFR may be due to the non-standard predictive methodology defined in Algorithm 2, suggesting potential refinements for future investigations.

### 5.3 LARGE-SCALE OCEAN MODELING

Lastly, we evaluate our methods on a real-world problem using public data sourced from the *AUSWAVE* physics model produced by the Australian Bureau of Meteorology [2016]. The task is to predict significant wave height across the spatial domain, shown in Figure 3, using $N = 5000$ randomly sampled locations and an input dimension $d = 8$ consisting of the spatial coordinates with additional physical model covariates.

We chose this setting as it is an inherently non-stationary domain, both due to the complexity of oceanographic modeling, as well as the fact that the distribution is not supported over the entire spatial domain. As such, we demonstrate the flexibility of M-SRFR to adapt to alternative kernel learning methodologies by introducing a non-stationary M-SRFR variant through the use of deep kernels [Wilson et al., 2016]. Specifically, we jointly train a neural network with 3 hidden layers and 32 activations per layer to first project the data before an M-SRFR kernel mixture of RFFs is applied. The neural network is trained jointly with the M-SRFR RFF parameters, but does not receive the same "mixture" treatment – ie. all $M$ M-SRFR kernels share the same input network.

In this sense, we are measuring M-SRFR's ability to slot in as a modular component to alternative kernel learning schemes, and whether the benefits of the mixture approach extend to such a setting. We include an SSGP with a deep kernel, SSGP-Deep, as an additional baseline in order to

differentiate the effect of the M-SRFR mixture from the neural network projection.

The results, shown in Figure 3 (right), show that M-SRFR's flexibility offers significant benefit. Most the of the stationary baselines, including traditional M-SRFR, have difficulty adjusting to the non-stationary domain. However, the deep M-SRFR variant significantly outperforms even the SSGP-Deep variant, as well as all baselines, demonstrating that there is unique value to the mixture approach. These results highlight that with little change of methodology, M-SRFR can be injected into alternative kernel learning schemes to improve performance and flexibility.

## 6 CONCLUSION

This study introduces Stein variational gradient descent to kernel approximation and Bayesian inference over spectral measures with random Fourier features and sparse spectrum Gaussian processes. We establish a theoretical framework linking these areas through functional inference, highlighting their coherence as particle-based methods. We derive a method for approximate inference of kernel spectral measures using only gradient evaluations of mixture components, which is straightforward to extend to many spectrum-based kernel learning methods. Empirical evaluations showcase the potential of integrating these methodologies to augment kernel approximation and sparse GP regression.

This work paves way for future research exploring the integration of RFFs and SVGD. One such avenue is the implementation of M-SRFR into other spectral kernel learning techniques of Section 3 to address challenges involving nonstationary processes, high-dimensional data, and non-Gaussian likelihoods. Future theoretical work will also focus on error and convergence analysis of our methods, which depend on further development of theoretical frameworks for the analysis of SVGD over infinite-dimensional spaces.

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

# Stein Random Feature Regression
# (Supplementary Material)

**Houston Warren**[1]  **Rafael Oliveira**[2]  **Fabio Ramos**[1,3]

[1]School of Computer Science, The University of Sydney, Sydney, Australia
[2]DATA61, CSIRO, Sydney, Australia,
[3]NVIDIA, USA,

## A  FUNCTIONAL KERNEL LEARNING WITH ENTROPY REGULARISATION

We can derive functional gradients from the definition of Fréchet derivatives and an analogous of Taylor's theorem for Hilbert spaces. Let $\mathcal{H}_\kappa^d := \bigotimes_{i=1}^d \mathcal{H}_\kappa$ be the vector-valued reproducing kernel Hilbert space defined by a positive semi-definite kernel $\kappa : \Omega \times \Omega \to \mathbb{R}$, for $\Omega \subseteq \mathbb{R}^d$, which has the following reproducing property:

$$\forall \mathbf{h} \in \mathcal{H}_\kappa^d, \quad \forall \mathbf{v} \in \mathbb{R}^d, \qquad \langle \mathbf{h}, \kappa(\cdot, \boldsymbol{\omega})\mathbf{v} \rangle_\kappa = \mathbf{h}(\boldsymbol{\omega})^\mathsf{T}\mathbf{v} \,, \tag{29}$$

where $\langle \cdot, \cdot \rangle_\kappa$ denotes the inner product in $\mathcal{H}_\kappa^d$. Following the SVGD setup, at each time step $t$, we consider a given variational probability distribution $q_t$ on $\mathbb{R}^d$ and apply a smooth transformation $T_{\mathbf{h}_t} : \mathbb{R}^d \to \mathbb{R}^d$ to its samples $\mathbf{x}_t \sim q_t$:

$$\boldsymbol{\omega}_{t+1} = T_{\mathbf{h}_t}(\boldsymbol{\omega}_t) = \boldsymbol{\omega}_t + \mathbf{h}_t(\boldsymbol{\omega}_t) \,, \tag{30}$$

where $\mathbf{h} \in \mathcal{H}_\kappa^d$, so that the next variational distribution is the pushforward of the former, i.e., $q_{t+1} := q_{\mathbf{h}_t} := T_{\mathbf{h}_t} \# q_t$.

We want to transform the GP kernel in its Fourier domain in order to minimise the GP negative log marginal likelihood. At the same time, to prevent overfitting, we may include an entropy-regularisation term into our objective, which allows for modeling uncertainty about the optimal $q$ when data is limited. This leads us to the following functional objective for the transform of the variational frequencies distribution $q_{\mathbf{h}} = T_{\mathbf{h}} \# q$:

$$F[\mathbf{h}] := L[q_{\mathbf{h}}] - \eta H[q_{\mathbf{h}}] \,, \tag{31}$$

for $\eta > 0$. Applying an SVGD-inspired approach, the optimal $q$ results from taking a sequence of optimal transformations in the RKHS $\mathcal{H}_\kappa^d$. The direction of steepest descent in $\mathcal{H}_\kappa^d$ is given by the functional gradient $\nabla_{\mathbf{h}} F[\mathbf{h}]$, which is such that:

**Definition 4** (Functional gradient). The gradient of a functional $F : \mathcal{H}^d \to \mathbb{R}$ at a point $\mathbf{h} \in \mathcal{H}^d$, defined over a Hilbert space $\mathcal{H}^d$ equipped with inner product $\langle \cdot, \cdot \rangle$, is the vector $\nabla_{\mathbf{h}} F[\mathbf{h}] \in \mathcal{H}^d$ such that:

$$F[\mathbf{h} + \epsilon \mathbf{g}] = F[\mathbf{h}] + \epsilon \langle \nabla_{\mathbf{h}} F[\mathbf{h}], g \rangle + \mathcal{O}(\epsilon^2) \,. \tag{32}$$

Given an initial $q$, applying an infinitesimal step in the direction of steepest descent means we only need to know $\nabla_{\mathbf{h}} F[\mathbf{h}]$ at the limit when $\mathbf{h} \to 0$.

### A.1  STATIONARY COVARIANCE FUNCTIONS

To calculate the functional gradient, we will follow the steps of Mallick et al. [2021] in the derivation of their functional gradient for the GP NLL. We start by noticing that, by the chain rule, we have:

$$\nabla_{\mathbf{h}} L[q_{\mathbf{h}}] = \sum_{i,j=1}^n \frac{\partial L}{\partial K_{ij}} \nabla_{\mathbf{h}} K_{ij}[\mathbf{h}] \,, \tag{33}$$

where:

$$K_{ij}[\mathbf{h}] := k_{\mathbf{h}}(\mathbf{x}_i, \mathbf{x}_j), \qquad i, j \in \{1, \ldots, n\} \tag{34}$$

$$k_{\mathbf{h}}(\mathbf{x}, \mathbf{x}') := \int_{\mathbb{R}^d} q_{\mathbf{h}}(\boldsymbol{\omega}) e^{\iota \boldsymbol{\omega}^{\mathsf{T}}(\mathbf{x}-\mathbf{x}')} \, \mathrm{d}\boldsymbol{\omega}, \quad \iota := \sqrt{-1} \tag{35}$$

Let $\rho_{ij}(\boldsymbol{\omega}) := e^{\iota \boldsymbol{\omega}^{\mathsf{T}}(\mathbf{x}_i - \mathbf{x}_j)}$, for $i, j \in \{1, \ldots, n\}$. Applying Taylor's expansion and the reproducing property leads us to:

$$
\begin{aligned}
\rho_{ij}(\boldsymbol{\omega} + \mathbf{h}(\boldsymbol{\omega}) + \epsilon \mathbf{g}(\boldsymbol{\omega})) &= \rho_{ij}(\boldsymbol{\omega} + \mathbf{h}(\boldsymbol{\omega})) + \epsilon \nabla \rho_{ij}(\boldsymbol{\omega} + \mathbf{h}(\boldsymbol{\omega})) \cdot \mathbf{g}(\boldsymbol{\omega}) + \mathcal{O}(\epsilon^2 \|\mathbf{g}(\boldsymbol{\omega})\|_2^2) \\
&= \rho_{ij}(\boldsymbol{\omega} + \mathbf{h}(\boldsymbol{\omega})) + \epsilon \langle \kappa(\cdot, \boldsymbol{\omega}) \nabla \rho_{ij}(\boldsymbol{\omega} + \mathbf{h}(\boldsymbol{\omega})), \mathbf{g} \rangle_{\kappa} + \mathcal{O}(\epsilon^2),
\end{aligned}
\tag{36}
$$

noting that $\mathcal{O}(\epsilon^2 \|\mathbf{g}(\boldsymbol{\omega})\|_2^2)$ is $\mathcal{O}(\epsilon^2)$, given that $\|\mathbf{g}(\boldsymbol{\omega})\|_2 \leq \|\mathbf{g}\|_{\kappa} \sqrt{\kappa(\boldsymbol{\omega}, \boldsymbol{\omega})}$ is bounded for any $\boldsymbol{\omega} \in \mathbb{R}^d$, assuming a bounded kernel. Since $K_{ij}[\mathbf{h}] = \mathbb{E}_{q_{\mathbf{h}}(\boldsymbol{\omega})}[\rho_{ij}(\boldsymbol{\omega})] = \mathbb{E}_{q(\boldsymbol{\omega})}[\rho_{ij}(\boldsymbol{\omega} + \mathbf{h}(\boldsymbol{\omega}))]$, we have that:

$$
\begin{aligned}
K_{ij}[\mathbf{h} + \epsilon \mathbf{g}] - K_{ij}[\mathbf{h}] &= \mathbb{E}_{q(\boldsymbol{\omega})}[\rho_{ij}(\boldsymbol{\omega} + \mathbf{h}(\boldsymbol{\omega}) + \epsilon \mathbf{g}(\boldsymbol{\omega})) - \rho_{ij}(\boldsymbol{\omega} + \mathbf{h}(\boldsymbol{\omega}))] \\
&= \epsilon \mathbb{E}_{q(\boldsymbol{\omega})}[\langle \kappa(\cdot, \boldsymbol{\omega}) \nabla \rho_{ij}(\boldsymbol{\omega} + \mathbf{h}(\boldsymbol{\omega})), \mathbf{g} \rangle_{\kappa}] + \mathcal{O}(\epsilon^2) \\
&= \epsilon \langle \mathbb{E}_{q(\boldsymbol{\omega})}[\kappa(\cdot, \boldsymbol{\omega}) \nabla \rho_{ij}(\boldsymbol{\omega} + \mathbf{h}(\boldsymbol{\omega}))], \mathbf{g} \rangle_{\kappa} + \mathcal{O}(\epsilon^2)
\end{aligned}
\tag{37}
$$

Applying Equation 32 to $K_{ij}[\mathbf{h}]$, the functional gradient of the kernel is then given by:

$$\nabla_{\mathbf{h}} K_{ij}[\mathbf{h}]\big|_{\mathbf{h}=\mathbf{0}} = \mathbb{E}_{q(\boldsymbol{\omega})}[\kappa(\cdot, \boldsymbol{\omega}) \nabla_{\boldsymbol{\omega}} \rho_{ij}(\boldsymbol{\omega})], \tag{38}$$

and, for the NLL, we have:

$$
\begin{aligned}
\nabla_{\mathbf{h}} L[q_{\mathbf{h}}]\big|_{\mathbf{h}=\mathbf{0}} &= \sum_{i,j=1}^{n} \frac{\partial L}{\partial K_{ij}} \mathbb{E}_{q(\boldsymbol{\omega})}[\kappa(\cdot, \boldsymbol{\omega}) \nabla_{\boldsymbol{\omega}} \rho_{ij}(\boldsymbol{\omega})] \\
&= \mathbb{E}_{q(\boldsymbol{\omega})} \left[ \kappa(\cdot, \boldsymbol{\omega}) \sum_{i,j=1}^{n} \frac{\partial L}{\partial K_{ij}} \nabla_{\boldsymbol{\omega}} \rho_{ij}(\boldsymbol{\omega}) \right].
\end{aligned}
\tag{39}
$$

For a particle-based approximation of the kernel $\hat{K}_{ij} \approx \frac{1}{R} \sum_{r=1}^{R} \cos(2\pi \boldsymbol{\omega}_r^{\mathsf{T}}(\mathbf{x}_i - \mathbf{x}_j))$, the equations above simplify to:

$$
\begin{aligned}
\nabla_{\mathbf{h}} K_{ij}[\mathbf{h}]\big|_{\mathbf{h}=\mathbf{0}} \approx \nabla_{\mathbf{h}} \hat{K}_{ij}[\mathbf{h}]\big|_{\mathbf{h}=\mathbf{0}} &= \frac{1}{R} \sum_{r=1}^{R} \kappa(\cdot, \boldsymbol{\omega}_r) \nabla_{\boldsymbol{\omega}_r} \cos(2\pi \boldsymbol{\omega}_r^{\mathsf{T}}(\mathbf{x}_i - \mathbf{x}_j)) \\
&= \sum_{r=1}^{R} \kappa(\cdot, \boldsymbol{\omega}_r) \nabla_{\boldsymbol{\omega}_r} \hat{K}_{ij},
\end{aligned}
\tag{40}
$$

and, for the NLL with the particle-based kernel, we have:

$$
\begin{aligned}
\nabla_{\mathbf{h}} L[q_{\mathbf{h}}]\big|_{\mathbf{h}=\mathbf{0}} \approx \nabla_{\mathbf{h}} \hat{L}[q_{\mathbf{h}}]\big|_{\mathbf{h}=\mathbf{0}} &= \sum_{r=1}^{R} \kappa(\cdot, \boldsymbol{\omega}_r) \sum_{i,j=1}^{n} \frac{\partial \hat{L}}{\partial \hat{K}_{ij}} \nabla_{\boldsymbol{\omega}_r} \hat{K}_{ij} \\
&= \sum_{r=1}^{R} \kappa(\cdot, \boldsymbol{\omega}_r) \nabla_{\boldsymbol{\omega}_r} \hat{L}[q],
\end{aligned}
\tag{41}
$$

which follows by another application of the chain rule.

Now, for the entropy-regularisation term, we have:

$$
\begin{aligned}
H[q_{\mathbf{h}}] &= \mathbb{E}_{q_{\mathbf{h}}(\boldsymbol{\omega})}[-\log q_{\mathbf{h}}(\boldsymbol{\omega})] \\
&= \mathbb{E}_{q(\boldsymbol{\omega})}[-\log q_{\mathbf{h}}(\boldsymbol{\omega} + \mathbf{h}(\boldsymbol{\omega}))]
\end{aligned}
\tag{42}
$$

By the change-of-variable formula for $q_{\mathbf{h}} = T_{\mathbf{h}} \# q$, we also have:

$$\log q_{\mathbf{h}}(\boldsymbol{\omega} + \mathbf{h}(\boldsymbol{\omega})) = \log q(\boldsymbol{\omega}) - \log|\det(\mathbf{I} + \nabla_{\boldsymbol{\omega}} \mathbf{h}(\boldsymbol{\omega}))|, \tag{43}$$

where $\nabla_{\boldsymbol{\omega}}\mathbf{h}(\boldsymbol{\omega})$ denotes the Jacobian matrix of $\mathbf{h}$. Applying the functional gradient formula (Equation 32) then yields:

$$
\begin{aligned}
H[q_{\mathbf{h}+\epsilon\mathbf{g}}] &= \mathbb{E}_{q_{\mathbf{h}+\epsilon\mathbf{g}}(\boldsymbol{\omega})}[-\log q_{\mathbf{h}+\epsilon\mathbf{g}}(\boldsymbol{\omega})] \\
&= \mathbb{E}_{q_{\mathbf{h}}(\boldsymbol{\omega})}[-\log q_{\mathbf{h}+\epsilon\mathbf{g}}(\boldsymbol{\omega}+\epsilon\mathbf{g}(\boldsymbol{\omega}))] \\
&= \mathbb{E}_{q_{\mathbf{h}}(\boldsymbol{\omega})}[-\log q_{\mathbf{h}}(\boldsymbol{\omega}) + \log|\det(\mathbf{I}+\epsilon\nabla_{\boldsymbol{\omega}}\mathbf{g}(\boldsymbol{\omega}))|] \\
&= H[q_{\mathbf{h}}] + \mathbb{E}_{q_{\mathbf{h}}(\boldsymbol{\omega})}[\log|\det(\mathbf{I}+\epsilon\nabla_{\boldsymbol{\omega}}\mathbf{g}(\boldsymbol{\omega}))|] \\
&= H[q_{\mathbf{h}}] + \epsilon\mathbb{E}_{q_{\mathbf{h}}(\boldsymbol{\omega})}\left[\mathrm{Tr}\left(\left.\frac{\partial\log|\det\mathbf{M}|}{\partial\mathbf{M}}\right|_{\mathbf{M}=\mathbf{I}}\nabla_{\boldsymbol{\omega}}\mathbf{g}(\boldsymbol{\omega})\right)\right] + \mathcal{O}(\epsilon^2) \\
&= H[q_{\mathbf{h}}] + \epsilon\mathbb{E}_{q_{\mathbf{h}}(\boldsymbol{\omega})}[\mathrm{Tr}\left(\nabla_{\boldsymbol{\omega}}\mathbf{g}(\boldsymbol{\omega})\right)] + \mathcal{O}(\epsilon^2) \\
&= H[q_{\mathbf{h}}] + \epsilon\mathbb{E}_{q_{\mathbf{h}}(\boldsymbol{\omega})}[\nabla_{\boldsymbol{\omega}}\cdot\mathbf{g}(\boldsymbol{\omega})] + \mathcal{O}(\epsilon^2) \\
&= H[q_{\mathbf{h}}] + \epsilon\langle\mathbb{E}_{q_{\mathbf{h}}(\boldsymbol{\omega})}[\nabla_{\boldsymbol{\omega}}\kappa(\cdot,\boldsymbol{\omega})],\mathbf{g}\rangle_{\kappa} + \mathcal{O}(\epsilon^2)\,,
\end{aligned}
\tag{44}
$$

where applied Taylor expansion to the log-determinant term around the identity matrix $\mathbf{I}$ and then the reproducing property of the kernel to extract the kernel gradient out of the divergent of $\mathbf{g}$. As a result, we have:

$$
\nabla_{\mathbf{h}}H[q_{\mathbf{h}}]\big|_{\mathbf{h}=\mathbf{0}} = \mathbb{E}_{q(\boldsymbol{\omega})}[\nabla_{\boldsymbol{\omega}}\kappa(\cdot,\boldsymbol{\omega})] \approx \frac{1}{R}\sum_{r=1}^{R}\nabla_{\boldsymbol{\omega}_r}\kappa(\cdot,\boldsymbol{\omega}_r)
\tag{45}
$$

Combining the results above finally yields:

$$
\begin{aligned}
\nabla_{\mathbf{h}}F[\mathbf{h}]\big|_{\mathbf{h}=\mathbf{0}} &= \mathbb{E}_{q(\boldsymbol{\omega})}\left[\kappa(\cdot,\boldsymbol{\omega})\sum_{i,j=1}^{n}\frac{\partial L}{\partial K_{ij}}\nabla_{\boldsymbol{\omega}}\rho_{ij}(\boldsymbol{\omega}) - \eta\nabla_{\boldsymbol{\omega}}\kappa(\cdot,\boldsymbol{\omega})\right] \\
&\approx \sum_{r=1}^{R}\kappa(\cdot,\boldsymbol{\omega}_r)\nabla_{\boldsymbol{\omega}_r}\hat{L}[q] - \frac{\eta}{R}\nabla_{\boldsymbol{\omega}_r}\kappa(\cdot,\boldsymbol{\omega}_r) \\
&= -\sum_{r=1}^{R}\kappa(\cdot,\boldsymbol{\omega}_r)\nabla_{\boldsymbol{\omega}_r}\log p(\mathbf{y}|\boldsymbol{\omega}_1,\ldots,\boldsymbol{\omega}_R) + \frac{\eta}{R}\nabla_{\boldsymbol{\omega}_r}\kappa(\cdot,\boldsymbol{\omega}_r)\,.
\end{aligned}
\tag{46}
$$

Note that the factor $\frac{1}{R}$ can further be absorbed into the regularisation factor $\eta$, which is a hyper-parameter of the algorithm.

**Frequency update steps.** Given the functional gradient formulation above, the resulting update steps are given by:

$$
\begin{aligned}
\boldsymbol{\omega}_i^{(t+1)} &= \boldsymbol{\omega}_i^{(t)} - \epsilon\nabla_{\mathbf{h}}F[\mathbf{h}](\boldsymbol{\omega}_i^{(t)})\big|_{\mathbf{h}=\mathbf{0}} \\
&\approx \boldsymbol{\omega}_i^{(t)} + \epsilon\sum_{r=1}^{R}\kappa(\boldsymbol{\omega}_i^{(t)},\boldsymbol{\omega}_r)\nabla_{\boldsymbol{\omega}_r}\log p(\mathbf{y}|\boldsymbol{\omega}_1,\ldots,\boldsymbol{\omega}_R) + \frac{\eta}{R}\nabla_{\boldsymbol{\omega}_r}\kappa(\boldsymbol{\omega}_i^{(t)},\boldsymbol{\omega}_r)\,,
\end{aligned}
\tag{47}
$$

for $i\in\{1,\ldots,R\}$.

# B  DISTRIBUTIONAL GRADIENT

In this section, we present a proof for Theorem 2 and a discussion on how we go from the theorem's result to operations over matrices of spectral frequencies.

## B.1  AUXILIARY NOTATION

We make use of notation shortcuts to express our main result in a compact form. We consider an RKHS $\mathcal{H}_{\boldsymbol{\Sigma}}$ of vector-valued functions associated with a positive-definite matrix-valued kernel $\boldsymbol{\Sigma}:\Omega\times\Omega\to\mathcal{B}(\Omega)$, where $\mathcal{B}(\Omega)$ denotes the space of bounded-linear operators mapping $\Omega$ to $\Omega$. When $\Omega\subseteq\mathbb{R}^d$, this simplifies to $\mathcal{B}(\Omega)=\mathbb{R}^{d\times d}$, i.e., the space of $d$-by-$d$ real-valued matrices. Furthermore, we will focus on the case of $\boldsymbol{\Sigma}(\boldsymbol{\omega},\boldsymbol{\omega}'):=\kappa(\boldsymbol{\omega},\boldsymbol{\omega}')\mathbf{I}$, where $\kappa:\Omega\times\Omega\to\mathbb{R}$ is positive-definite scalar-valued kernel, for all $\boldsymbol{\omega},\boldsymbol{\omega}'\in\Omega$. In this case, it is not hard to show that $\mathcal{H}_{\boldsymbol{\Sigma}}=\mathcal{H}_{\kappa}^d:=\bigotimes_{i=1}^{d}\mathcal{H}_{\kappa}$, i.e., the Cartesian product of $d$ copies of the scalar-valued RKHS $\mathcal{H}_{\kappa}$.

**Point evaluation and inner product.** The reproducing property of a kernel $\boldsymbol{\Sigma}$ associated with a vector-valued RKHS states that [Alvarez et al., 2012]:

$$\forall \mathbf{s} \in \mathcal{H}_{\boldsymbol{\Sigma}}, \quad \langle \mathbf{s}, \boldsymbol{\Sigma}(\cdot, \boldsymbol{\omega})\mathbf{v}\rangle_{\boldsymbol{\Sigma}} = \langle \mathbf{s}(\boldsymbol{\omega}), \mathbf{v}\rangle_2 = \mathbf{s}(\boldsymbol{\omega})^{\mathsf{T}}\mathbf{v}, \forall \mathbf{v} \in \mathbb{R}^d, \tag{48}$$

where $\langle \cdot, \cdot \rangle_2$ denotes the inner product associated with the 2-norm, i.e., the dot product in this case. If $\boldsymbol{\Sigma}(\boldsymbol{\omega}, \boldsymbol{\omega}') := \kappa(\boldsymbol{\omega}, \boldsymbol{\omega}')\mathbf{I}$, $\forall \boldsymbol{\omega}, \boldsymbol{\omega}' \in \Omega$, we then have that $\mathbf{s}(\cdot)^{\mathsf{T}}\mathbf{v} : \Omega \to \mathbb{R}$ is an element of $\mathcal{H}_{\kappa}$, so that $\mathbf{s}(\boldsymbol{\omega})^{\mathsf{T}}\mathbf{v} = \langle \mathbf{s}(\cdot)^{\mathsf{T}}\mathbf{v}, \kappa(\cdot, \boldsymbol{\omega})\rangle_{\kappa}$. Therefore, we will denote inner products in this vector-valued RKHS with the same notation subscript as for the scalar-valued case:

$$\langle \mathbf{s}, \kappa(\cdot, \boldsymbol{\omega})\mathbf{v}\rangle_{\kappa} := \langle \mathbf{s}, \kappa(\cdot, \boldsymbol{\omega})\mathbf{v}\rangle_{\boldsymbol{\Sigma}} = \langle \mathbf{s}, \boldsymbol{\Sigma}(\cdot, \boldsymbol{\omega})\mathbf{v}\rangle_{\boldsymbol{\Sigma}}. \tag{49}$$

**Matrix-valued evaluations.** Evaluating a function $\mathbf{s} \in \mathcal{H}_{\kappa}^d$ on a matrix of inputs $\boldsymbol{\Omega}_n := [\boldsymbol{\omega}_1, \ldots, \boldsymbol{\omega}_n]^{\mathsf{T}} \in \mathbb{R}^{n \times d}$ yields $\mathbf{s}(\boldsymbol{\Omega}_n) := [\mathbf{s}(\boldsymbol{\omega}_1), \ldots, \mathbf{s}(\boldsymbol{\omega}_n)]^{\mathsf{T}} \in \mathbb{R}^{n \times d}$. By the reproducing property of the kernel, we also have that:

$$\forall \mathbf{M} := [\mathbf{m}_1, \ldots, \mathbf{m}_n]^{\mathsf{T}} \in \mathbb{R}^{n \times d}, \quad \langle \mathbf{s}(\boldsymbol{\Omega}_n), \mathbf{M}\rangle_2 = \mathrm{Tr}(\mathbf{s}(\boldsymbol{\Omega}_n)^{\mathsf{T}}\mathbf{M})$$

$$= \mathrm{Tr}\left(\sum_{i=1}^n \mathbf{s}(\boldsymbol{\omega}_i)\mathbf{m}_i^{\mathsf{T}}\right)$$

$$= \sum_{i=1}^n \mathbf{s}(\boldsymbol{\omega}_i)^{\mathsf{T}}\mathbf{m}_i \tag{50}$$

$$= \sum_{i=1}^n \langle \mathbf{s}, \kappa(\cdot, \boldsymbol{\omega}_i)\mathbf{m}_i\rangle_{\kappa}$$

$$= \left\langle \mathbf{s}, \sum_{i=1}^n \kappa(\cdot, \boldsymbol{\omega}_i)\mathbf{m}_i\right\rangle_{\kappa},$$

where $\langle \cdot, \cdot \rangle_2$ corresponds to the Frobenius inner product when applied to matrices. We therefore denote $\kappa(\cdot, \boldsymbol{\Omega}_n)$ as the operator mapping matrices in $\mathbb{R}^{n \times d}$ to functions in $\mathcal{H}_{\kappa}^d$ which is such that:

$$\kappa(\cdot, \boldsymbol{\Omega}_n)\mathbf{M} = \sum_{i=1}^n \kappa(\cdot, \boldsymbol{\omega}_i)\mathbf{m}_i \in \mathcal{H}_{\kappa}^d \tag{51}$$

$$\kappa(\boldsymbol{\Omega}_m', \boldsymbol{\Omega}_n)\mathbf{M} = \begin{bmatrix} \kappa(\boldsymbol{\omega}_1', \boldsymbol{\omega}_1) & \cdots & \kappa(\boldsymbol{\omega}_1', \boldsymbol{\omega}_n) \\ \vdots & \ddots & \vdots \\ \kappa(\boldsymbol{\omega}_m', \boldsymbol{\omega}_1) & \cdots & \kappa(\boldsymbol{\omega}_m', \boldsymbol{\omega}_n) \end{bmatrix} \mathbf{M} \tag{52}$$

for any $\boldsymbol{\Omega}_m' := [\boldsymbol{\omega}_1', \ldots, \boldsymbol{\omega}_m']^{\mathsf{T}} \in \mathbb{R}^{m \times d}$.

**Jacobians.** The reproducing property also allows us to express Jacobians of a function in terms of kernel gradients as:

$$\forall \mathbf{s} \in \mathcal{H}_{\kappa}^d, \quad \nabla_{\mathbf{v}}\mathbf{s}(\mathbf{v}) = \langle \mathbf{s}, \nabla_{\mathbf{v}}\kappa(\cdot, \mathbf{v})\rangle_{\kappa}, \quad \forall \mathbf{v} \in \mathbb{R}^d. \tag{53}$$

For matrix-valued transformations $\boldsymbol{\Omega}_n \mapsto \mathbf{s}(\boldsymbol{\Omega}_n)$ [Gupta and Nagar, 1999, Ch. 1], we further have that:

$$\nabla_{\boldsymbol{\Omega}_n}\mathbf{s}(\boldsymbol{\Omega}_n) := \begin{bmatrix} \nabla_{\boldsymbol{\omega}_1}\mathbf{s}(\boldsymbol{\omega}_1) & \cdots & \nabla_{\boldsymbol{\omega}_1}\mathbf{s}(\boldsymbol{\omega}_n) \\ \vdots & \ddots & \vdots \\ \nabla_{\boldsymbol{\omega}_n}\mathbf{s}(\boldsymbol{\omega}_1) & \cdots & \nabla_{\boldsymbol{\omega}_n}\mathbf{s}(\boldsymbol{\omega}_n) \end{bmatrix} = \begin{bmatrix} \nabla_{\boldsymbol{\omega}_1}\mathbf{s}(\boldsymbol{\omega}_1) & \mathbf{0} & \cdots & \mathbf{0} \\ \mathbf{0} & \nabla_{\boldsymbol{\omega}_2}\mathbf{s}(\boldsymbol{\omega}_2) & \cdots & \mathbf{0} \\ \vdots & \vdots & \ddots & \vdots \\ \mathbf{0} & \mathbf{0} & \cdots & \nabla_{\boldsymbol{\omega}_n}\mathbf{s}(\boldsymbol{\omega}_n) \end{bmatrix}, \tag{54}$$

so that the following holds for the trace and the determinant of the Jacobian:

$$\mathrm{Tr}(\nabla_{\boldsymbol{\Omega}_n}\mathbf{s}(\boldsymbol{\Omega}_n)) = \sum_{i=1}^n \mathrm{Tr}(\nabla_{\boldsymbol{\omega}_i}\mathbf{s}(\boldsymbol{\omega}_i)) \tag{55}$$

$$|\nabla_{\boldsymbol{\Omega}_n}\mathbf{s}(\boldsymbol{\Omega}_n)| = \prod_{i=1}^n |\nabla_{\boldsymbol{\omega}_i}\mathbf{s}(\boldsymbol{\omega}_i)|, \tag{56}$$

where $|\cdot|$ denotes the absolute value of the determinant. Considering the reproducing property of the kernel, we have that:

$$\mathrm{Tr}(\nabla_{\boldsymbol{\Omega}_n}\mathbf{s}(\boldsymbol{\Omega}_n)) = \sum_{i=1}^n \mathrm{Tr}(\nabla_{\boldsymbol{\omega}_i}\mathbf{s}(\boldsymbol{\omega}_i)) = \left\langle \mathbf{s}, \sum_{i=1}^n \nabla_{\boldsymbol{\omega}_i}\kappa(\cdot, \boldsymbol{\omega}_i)\right\rangle_{\kappa} = \langle \mathbf{s}, \nabla_{\boldsymbol{\Omega}_n}\kappa(\cdot, \boldsymbol{\Omega}_n)\rangle_{\kappa}. \tag{57}$$

## B.2 PROOF OF MAIN RESULT

*Proof of Theorem 2.* Let $\mathbf{g}, \mathbf{s} \in \mathcal{H}_\kappa^d$, where $\kappa : \mathbb{R}^d \times \mathbb{R}^d \to \mathbb{R}$ is a positive-definite kernel over $\mathbb{R}^d$. Define a transform $T_\mathbf{g} : \mathcal{F} \to \mathcal{F}$ as a mapping such that $T_\mathbf{g}(\mathbf{h})(\boldsymbol{\omega}) = \mathbf{h}(\boldsymbol{\omega}) + \mathbf{g}(\mathbf{h}(\boldsymbol{\omega}))$, for all $\boldsymbol{\omega} \in \mathbb{R}^d$. Considering the definition of functional gradient (Definition 4) and the KL divergence between the measures $Q_\mathbf{g} := T_\mathbf{g} \# Q$ and $P_*$, we have:

$$
\begin{aligned}
D_{\mathrm{KL}}(Q_{\mathbf{g}+\epsilon\mathbf{s}}||P_*) &= \sup_{n,\boldsymbol{\Omega}_n} D_{\mathrm{KL}}(q_{\mathbf{g}+\epsilon\mathbf{s}}(\mathbf{H}_n)||p_*(\mathbf{H}_n)) \\
&= D_{\mathrm{KL}}(q_{\mathbf{g}+\epsilon\mathbf{s}}(\mathbf{H}^*)||p_*(\mathbf{H}^*)) \\
&= \mathbb{E}_{\mathbf{h}\sim Q_\mathbf{g}}\left[\log q_{\mathbf{g}+\epsilon\mathbf{s}}(\mathbf{H}^* + \epsilon\mathbf{s}(\mathbf{H}^*)) - \log p_*(\mathbf{H}^* + \epsilon\mathbf{s}(\mathbf{H}^*))\right],
\end{aligned} \tag{58}
$$

where $\mathbf{H}^* := \mathbf{h}(\boldsymbol{\Omega}_{n^*}^*)$, as defined in the theorem statement, assuming the supremum is achieved at a finite $n^*$ and that $\boldsymbol{\Omega}_{n^*}^*$ exists. Now applying the change-of-variable formula and a Taylor expansion on the resulting log-determinant, for the first term in the supremum, taking any $n \in \mathbb{N}$ and $\boldsymbol{\Omega}_n \subset \Omega$, with $\mathbf{H}_n := \mathbf{h}(\boldsymbol{\Omega}_n)$, $\mathbf{h} \sim Q_\mathbf{g}$, we have:

$$
\begin{aligned}
\log q_{\mathbf{g}+\epsilon\mathbf{s}}(\mathbf{H}_n + \epsilon\mathbf{s}(\mathbf{H}_n)) &= \log q_\mathbf{g}(\mathbf{H}_n) - \log|\mathbf{I} + \epsilon\nabla_{\mathbf{H}_n}\mathbf{s}(\mathbf{H}_n)| \\
&= \log q_\mathbf{g}(\mathbf{H}_n) - \log|\mathbf{I}| - \epsilon\left\langle \nabla_\mathbf{M}\log|\mathbf{M}|\Big|_{\mathbf{M}=\mathbf{I}}, \nabla_{\mathbf{H}_n}\mathbf{s}(\mathbf{H}_n)\right\rangle_2 + \mathcal{O}(\epsilon^2) \\
&= \log q_\mathbf{g}(\mathbf{H}_n) - \epsilon\mathrm{Tr}(\nabla_{\mathbf{H}_n}\mathbf{s}(\mathbf{H}_n)) + \mathcal{O}(\epsilon^2) \\
&= \log q_\mathbf{g}(\mathbf{H}_n) - \epsilon\langle\mathbf{s}, \nabla_{\mathbf{H}_n}\kappa(\cdot, \mathbf{H}_n)\rangle_\kappa + \mathcal{O}(\epsilon^2)
\end{aligned} \tag{59}
$$

where $\langle\cdot,\cdot\rangle_2$ here denotes the Frobenius inner product between matrices, and we applied the reproducing property of $\kappa$ to derive the last term.

For the second term in the supremum, also applying Taylor's theorem and the reproducing property yields:

$$
\begin{aligned}
\log p_*(\mathbf{H}_n + \epsilon\mathbf{s}(\mathbf{H}_n)) &= \log p_*(\mathbf{H}_n) + \epsilon\langle\nabla_{\mathbf{H}_n}\log p(\mathbf{H}_n), \mathbf{s}(\mathbf{H}_n)\rangle_2 + \mathcal{O}(\epsilon^2) \\
&= \log p_*(\mathbf{H}_n) + \epsilon\langle\kappa(\cdot, \mathbf{H}_n)\nabla_{\mathbf{H}_n}\log p(\mathbf{H}_n), \mathbf{s}\rangle_\kappa + \mathcal{O}(\epsilon^2).
\end{aligned} \tag{60}
$$

Combining the two equations above into the KL divergence, and applying the definition of functional gradient leads us to:

$$
\nabla_\mathbf{g} D_{\mathrm{KL}}(Q_\mathbf{g}||P_*) = -\mathbb{E}_{\mathbf{h}\sim Q_\mathbf{g}}[\kappa(\cdot, \mathbf{H}^*)\nabla_{\mathbf{H}^*}\log p(\mathbf{H}^*) + \nabla_{\mathbf{H}^*}\kappa(\cdot, \mathbf{H}^*)], \tag{61}
$$

which yields the first result in Theorem 2 by letting $\mathbf{g} \to 0$.

When applied to transforms over a finite set $\boldsymbol{\Omega}_R = \{\boldsymbol{\omega}_i\}_{i=1}^R \subset \mathbb{R}^d$, the KL divergence simplifies to:

$$
D_{\mathrm{KL}}(Q||P^*) = \sup_{n\leq R, \boldsymbol{\Omega}_n \subset \boldsymbol{\Omega}_R} D_{\mathrm{KL}}(q(\mathbf{h}(\boldsymbol{\Omega}_n))||p_*(\mathbf{h}(\boldsymbol{\Omega}_n))) = D_{\mathrm{KL}}(q(\mathbf{h}(\boldsymbol{\Omega}_R))||p_*(\mathbf{h}(\boldsymbol{\Omega}_R))). \tag{62}
$$

The second result in Theorem 2 then arises by setting $\mathbf{H}^* := \mathbf{h}(\boldsymbol{\Omega}_R)$ in Equation 61 and letting $\mathbf{g} \to 0$. $\qquad\square$

## B.3 FROM STOCHASTIC PROCESSES TO DISTRIBUTIONS OVER MATRICES

As a note, we here discuss how Theorem 2 gives rise to our algorithmic setting, which operates over matrices of frequencies, representing empirical spectral measures. For any fixed $\boldsymbol{\Omega}_R \in \mathbb{R}^{R\times d}$ and i.i.d. samples $\{\mathbf{h}_i\}_{i=1}^M \overset{i.i.d.}{\sim} Q$, note that $\boldsymbol{\Omega}_R^{(i)} := \mathbf{h}_i(\boldsymbol{\Omega}_R)$ corresponds to the realisation of a random matrix. The corresponding matrix distribution is given by $q_R := \mathrm{E}_{\boldsymbol{\Omega}_R} \# Q$, where $\mathrm{E}_{\boldsymbol{\Omega}_R}$ is the matrix-valued evaluation operator defined as:

$$
\begin{aligned}
\mathrm{E}_{\boldsymbol{\Omega}_R} : \mathcal{H}_\kappa^d &\to \mathbb{R}^{R\times d} \\
\mathbf{h} &\mapsto \mathbf{h}(\boldsymbol{\Omega}_R),
\end{aligned} \tag{63}
$$

recalling that $\mathbf{h}(\boldsymbol{\Omega}_R) := [\mathbf{h}(\boldsymbol{\omega}_1), \ldots, \mathbf{h}(\boldsymbol{\omega}_R)]^\mathsf{T} \in \mathbb{R}^{R\times d}$. Therefore, we can rewrite an expectation over $Q$ as an expectation over $q_R$ when it takes the following form:

$$
\mathbb{E}_{\mathbf{h}\sim Q}[f(\mathbf{h}(\boldsymbol{\Omega}_R))] = \mathbb{E}_{\boldsymbol{\Omega}\sim q_R}[f(\boldsymbol{\Omega})], \tag{64}
$$

for any integrable $f : \mathbb{R}^{R\times d} \to \mathbb{R}$. The initial distribution is arbitrary, and subsequent SVGD steps operate directly on the samples. Hence, we can replace $\mathbf{H}$ in Theorem 2 with the matrix particles in the empirical approximations used by SVGD.

### B.4 SVGD UPDATE

Theorem 2 results in the following SVGD $M$-particle update rule:

$$\boldsymbol{\Omega}_m^{t+1} = \boldsymbol{\Omega}_m^t + \frac{\epsilon}{M} \sum_{j=1}^{M} \kappa(\boldsymbol{\Omega}_m^t, \boldsymbol{\Omega}_j^t) \nabla_{\boldsymbol{\Omega}_j^t} \log p(\boldsymbol{\Omega}_j^t) + \nabla_{\boldsymbol{\Omega}_j^t} \kappa(\boldsymbol{\Omega}_m^t, \boldsymbol{\Omega}_j^t), \tag{65}$$

where, according to the notation for Theorem 2, for any $\boldsymbol{\Omega}, \boldsymbol{\Omega}' \in \mathbb{R}^{R \times d}$, we have the kernel matrix as:

$$\kappa(\boldsymbol{\Omega}, \boldsymbol{\Omega}') = \begin{bmatrix} \kappa(\boldsymbol{\omega}_1, \boldsymbol{\omega}_1') & \dots & \kappa(\boldsymbol{\omega}_1, \boldsymbol{\omega}_R') \\ \vdots & \ddots & \vdots \\ \kappa(\boldsymbol{\omega}_R, \boldsymbol{\omega}_1') & \dots & \kappa(\boldsymbol{\omega}_R, \boldsymbol{\omega}_R') \end{bmatrix} \in \mathbb{R}^{R \times R}, \tag{66}$$

and the kernel matrix-valued gradient is given by:

$$\nabla_{\boldsymbol{\Omega}'} \kappa(\boldsymbol{\Omega}, \boldsymbol{\Omega}') = \begin{bmatrix} \sum_{j=1}^{R} \nabla_{\boldsymbol{\omega}_j'} \kappa(\boldsymbol{\omega}_1, \boldsymbol{\omega}_j')^{\mathsf{T}} \\ \vdots \\ \sum_{j=1}^{R} \nabla_{\boldsymbol{\omega}_j'} \kappa(\boldsymbol{\omega}_R, \boldsymbol{\omega}_j')^{\mathsf{T}} \end{bmatrix} = \sum_{j=1}^{R} \nabla_{\boldsymbol{\omega}_j'} \begin{pmatrix} \kappa(\boldsymbol{\omega}_1, \boldsymbol{\omega}_j') \\ \vdots \\ \kappa(\boldsymbol{\omega}_R, \boldsymbol{\omega}_j') \end{pmatrix} = \sum_{j=1}^{R} \nabla_{\boldsymbol{\omega}_j'} \kappa(\boldsymbol{\Omega}, \boldsymbol{\omega}_j') \in \mathbb{R}^{R \times d}, \tag{67}$$

which is the sum of Jacobian matrices of the vector-valued map $\kappa(\boldsymbol{\Omega}, \cdot) : \boldsymbol{\omega} \mapsto \kappa(\boldsymbol{\Omega}, \boldsymbol{\omega}) := [\kappa(\boldsymbol{\omega}_1, \boldsymbol{\omega}) \dots \kappa(\boldsymbol{\omega}_R, \boldsymbol{\omega})]^{\mathsf{T}} \in \mathbb{R}^R$.

## C EXPERIMENTAL DETAILS

All experiments were performed on a single desktop using a AMD Ryzen 7 5800X CPU, 32GB of RAM, and an NVIDIA 3080 Ti GPU. Dataset sizes are as follows:

| Dataset | $N$ | $d$ | $N_{train}$ | $N_{test}$ |
|---------|-----|-----|-------------|------------|
| airfoil | 1503 | 5 | 1353 | 150 |
| concrete | 1030 | 8 | 824 | 206 |
| energy | 768 | 16 | 615 | 153 |
| wine | 1599 | 11 | 1440 | 159 |
| AUSWAVE | 1787594 | 8 | 5000 | 1000 |

### C.1 HYPERPARAMETER TRAINING

We perform the following number of hyperparameter runs for all experiments, increasing the number of runs as hyperparameter count grows

| Model | Runs | # of Hyperparameters |
|-------|------|----------------------|
| SVGP | 30 | 1 |
| SSGP-RBF | 30 | 2 |
| SSGP/SSGP-$R^*$ | 30 | 2 |
| SSGP-SVGD | 50 | 5 |
| M-SRFR | 75 | 6 |

## D VISUALIZATIONS OF MIXTURE KERNELS AND PREDICTIONS

We provide a visualization of the learned M-SRFR kernels, as well as a selection of the predictive mixture and combined distributions, for the UCI datasets.

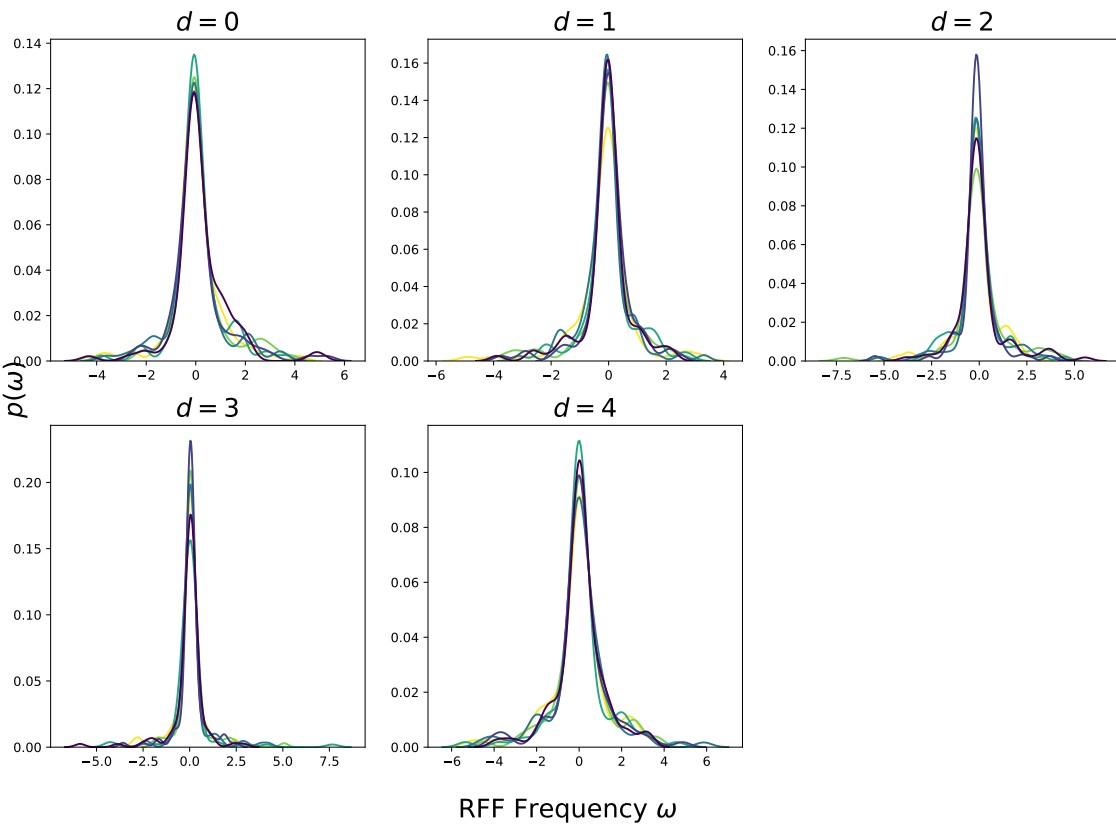

Figure 4: *airfoil* Learned Kernels by Dimension.

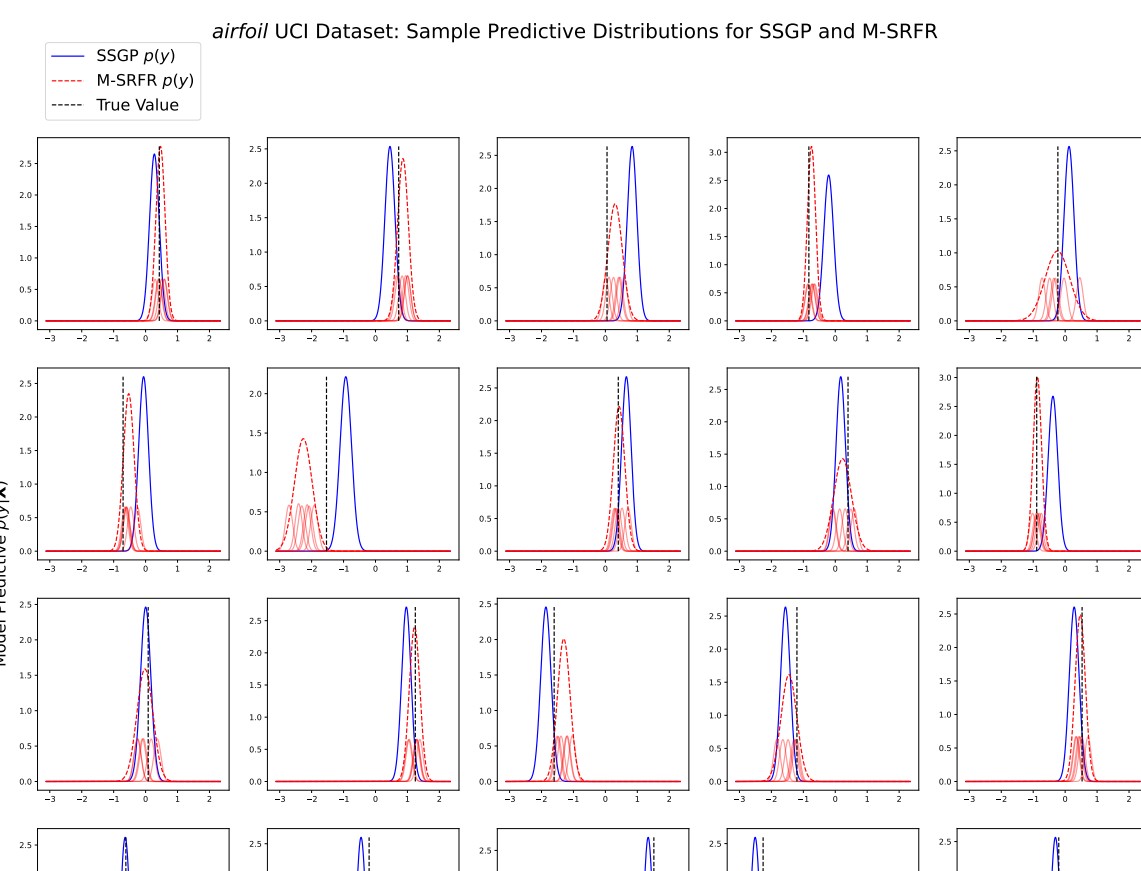

Figure 5: Selection of Single SSGP and M-SRFR Predictive Distributions for *airfoil* Test Points.

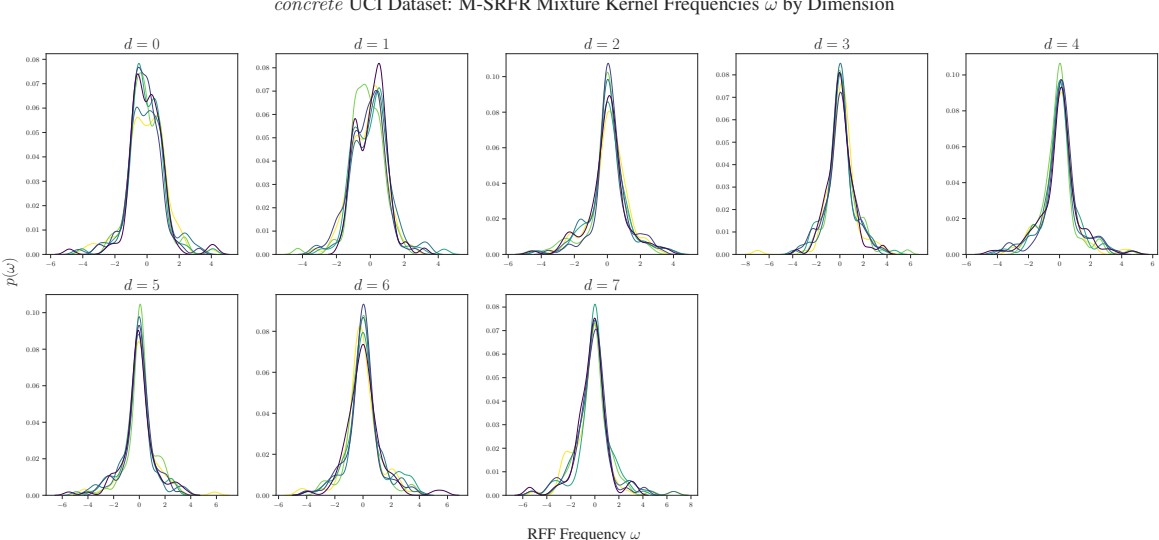

Figure 6: *concrete* Learned Kernels by Dimension.

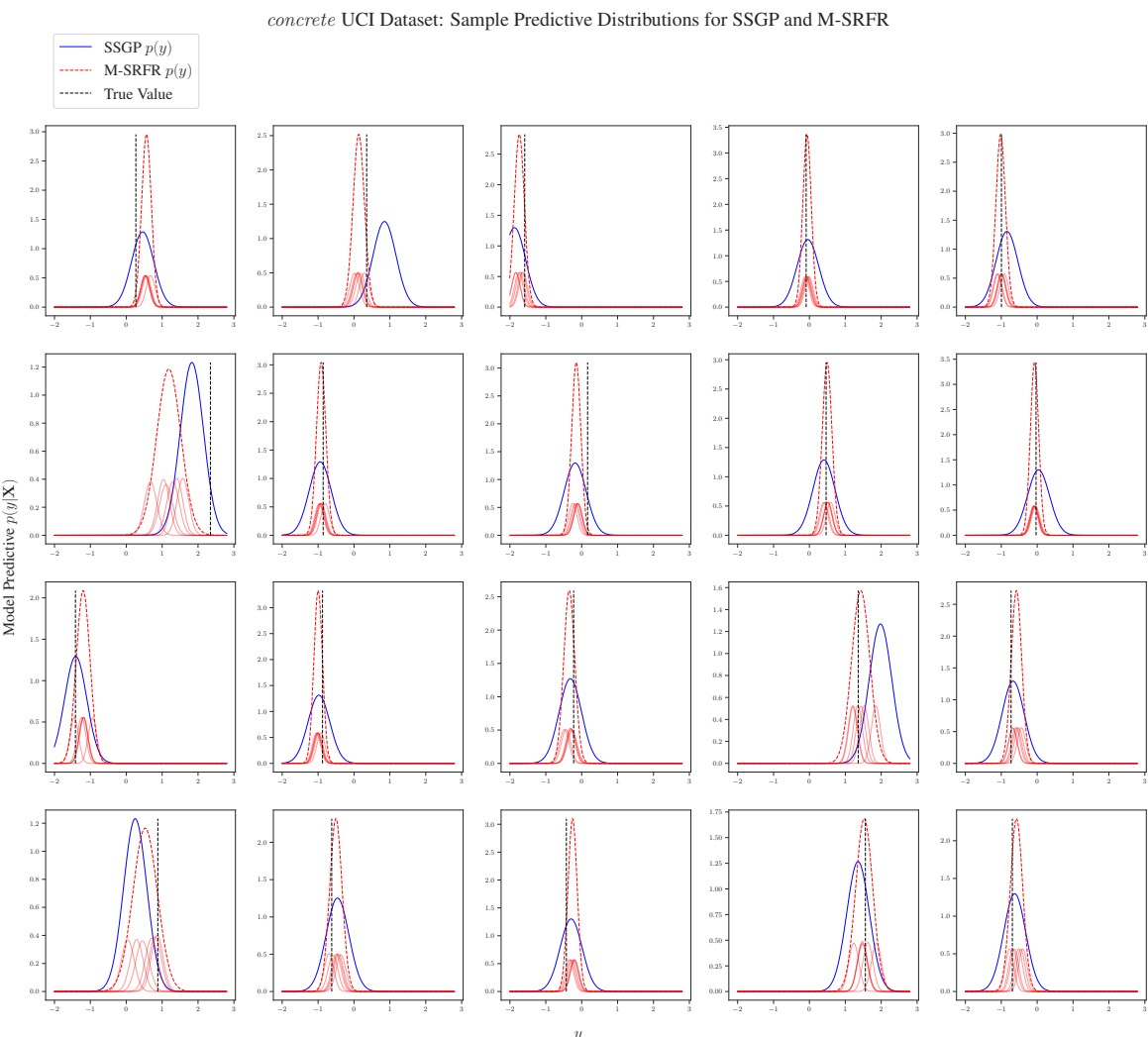

Figure 7: Selection of Single SSGP and M-SRFR Predictive Distributions for *concrete* Test Points.

*energy* UCI Dataset: M-SRFR Mixture Kernel Frequencies $\omega$ by Dimension

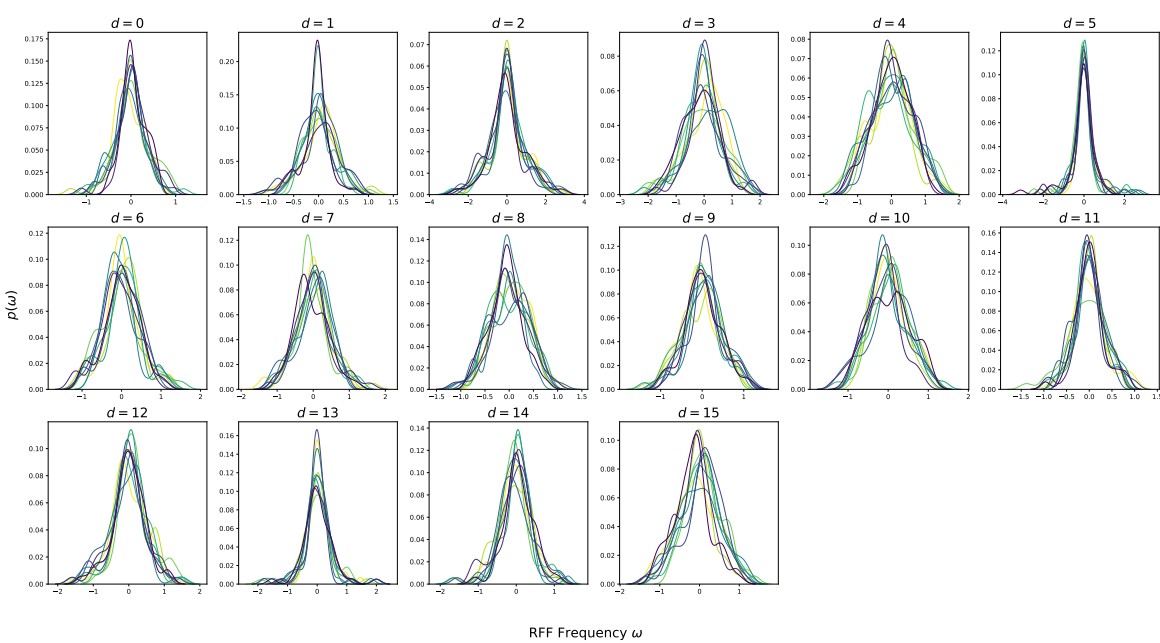

Figure 8: *energy* Learned Kernels by Dimension.

energy UCI Dataset: Sample Predictive Distributions for SSGP and M-SRFR

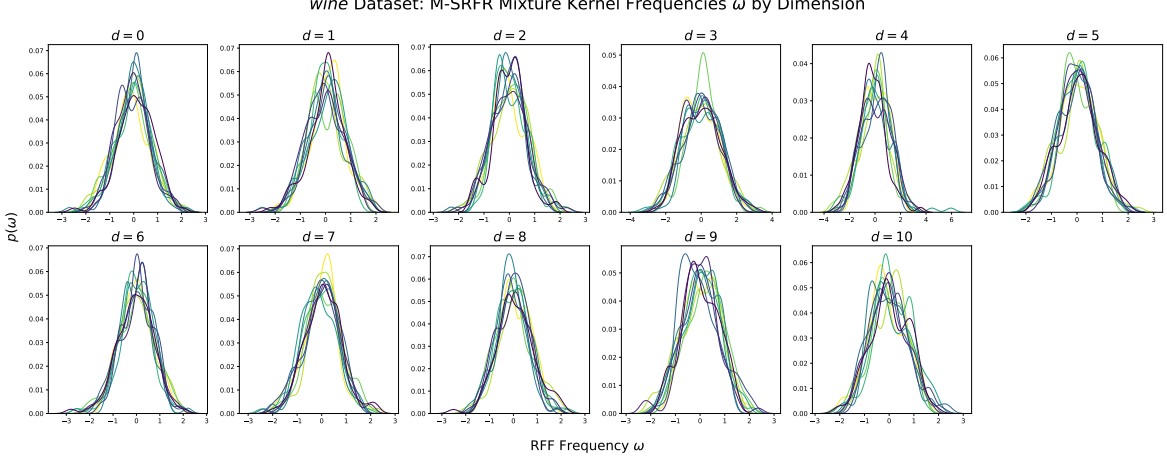

Figure 9: Selection of Single SSGP and M-SRFR Predictive Distributions for *energy* Test Points.

wine Dataset: M-SRFR Mixture Kernel Frequencies $\omega$ by Dimension

Figure 10: *wine* Learned Kernels by Dimension.

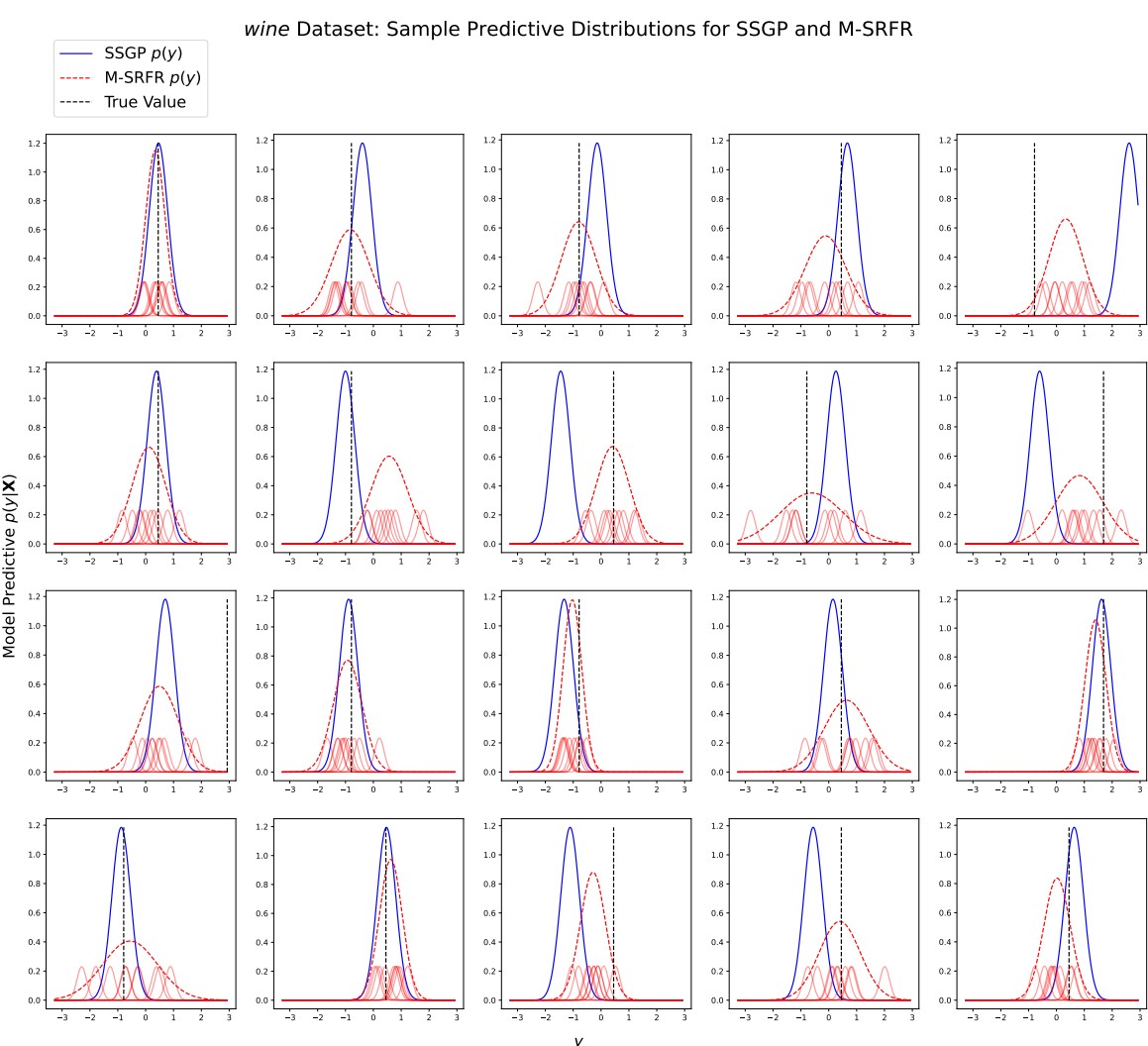

Figure 11: Selection of Single SSGP and M-SRFR Predictive Distributions for *wine* Test Points.