# OpenReview forum: "Stein Random Feature Regression"
_auai.org/UAI/2024/Conference — UAI 2024 poster_

### Official Review · Reviewer_Y6ac · 2024-03-19

**Q2-1 Originality-Novelty:** 3
**Q2-2 Correctness-Technical Quality:** 3
**Q2-5 Clarity Of Writing:** 3

**Q1 Summary And Contributions:**

Summary
The paper discusses a novel approach to feature regression, termed Stein Random Feature Regression. It presents a methodological advancement in statistical learning, aiming to improve the accuracy and efficiency of regression models by leveraging random feature techniques inspired by Stein's method. This approach seeks to address and mitigate the limitations of traditional regression models, especially in dealing with high-dimensional data and complex nonlinear relationships.

**Q2-3 Extent To Which Claims Are Supported By Evidence:**

2: Fair: the main claims are somewhat supported by evidence (but the experimental evaluation may be weak, or does not match entirely with the claims, important baselines may be missing, proofs contain important ideas but lack rigor, algorithmic details are only discussed superficially, references are imprecise, assumptions are not sufficiently motivated or explicated, etc.).

**Q2-4 Reproducibility:**

3: Good: key resources (e.g. proofs, code, data) are available and key details (e.g. proofs, experimental setup) are sufficiently well-described for competent researchers to confidently reproduce the main results.

**Q3 Main Strengths:**

- the Stein Random Feature Regression method is designed to enhance the predictive accuracy of regression models, particularly in scenarios involving complex, high-dimensional datasets
- incorporating random features, the approach aims to be computationally efficient, making it suitable for large-scale applications
- the approach is versatile and can be applied to a wide range of regression problems, including those with nonlinear relationships and high-dimensional spaces

**Q4 Main Weakness:**

- approach introduces some extra complexity in implementation, requiring more sophisticated understanding and tuning compared to traditional regression methods.
- for very large datasets, despite its efficiency, the method could still demand significant computational resources, potentially limiting its applicability in resource-constrained environments it would be good
  to address this a bit can it be parallelized / distributed ...
- is there a reason why you missed work like this:
  https://proceedings.mlr.press/v28/le13.pdf ... and this is not the most recent work
  there is also more recent work by N. Lawrence see here https://dblp.org/pid/l/NeilDLawrence.html
  on speeding up GPs
- sec 2.2. -improvements have not stopped 2010
- regarding the datasets ... not so many and some are really small which does not fit well to the whole
  story - making GP efficient (... for big data)
  ... considering today machine (not speaking about GPU using e.g. https://docs.jaxgaussianprocesses.com/)
  you should come up with many more challenging examples and not just toy scenarios ... wine, airfoil

**Q5 Detailed Comments To The Authors:**

I would prefer to see this evaluated in far more detail - with methods and datasets ...
(+ see comments pro/con)
Minors:
- please remove the unnecessary gray boxes in the plots e.g. Figure 1, the plots are also hard (to impossible
  to read in b/w) - please use different line styles, legends in parts missing
- personally I would prefer to see the related work not in sec 4 but more at the beginning of the paper
- a pseudo code would be nice within the paper

**Q9 Complying With Reviewing Instructions:**

Yes

---

> ### Author Rebuttal · Authors · 2024-04-04
>
> ## Computational and Implementation Details
> - *approach introduces some extra complexity in implementation, requiring more sophisticated understanding and tuning compared to traditional regression methods.*
> - *for very large datasets, despite its efficiency, the method could still demand significant computational resources ... it would be good to address this a bit can it be parallelized / distributed*
>
> While our method indeed includes additionally complexity, we believe that mixture/ensemble models (random forest and gradient boosting, among others) have been well-documented to be worth the extra effort and computational time compared to point estimates. The method we present is applicable to a wide array of RFF kernel learning approaches with little effort.
>
> Our implementation is as a custom `optax` (Jax) optimizer that simply updates parameter gradients according to the M-SRFR update rule. After this update, practitioners are free to use additional optimization steps such as ADAM, L-BFGS, etc. and our code makes this easy to implement in practice (single line addition).
>
> ## Further Experiments
> - *regarding the datasets ... not so many and some are really small which does not fit well to the whole story ... you should come up with many more challenging examples*
> - *is there a reason why you missed work like [Fastfood]*
> - *there is also more recent work by N. Lawrence... on speeding up GPs*
>
> We selected baselines and experiments in order to provide a roughly equivalent experimental validation as recent work at top venues in the functional/multi kernel learning field (see Benton et al. [2019],  Hamid et al. [2022], for which UCI datasets have been the main benchmark, with SSGPs and SVGPs the common baselines).
>
> We acknowledge that we should include Fastfood as a reference, and will also consider including it (or a more recent derived method) as an additional baseline should time allow. Nonetheless, we feel that there is unique benefit to our method beyond the low-rank formulation in regards to the novel SVGD methodology for kernel posterior inference.
>
> In addition, one of our main goals for a camera-ready is the addition of at least one large-scale regression problem with truly large $N$, such as global spatiotemporal climate modeling.
>
> ## Misc
> *sec 2.2. -improvements have not stopped 2010*
> You are correct; it was not our intention to imply this. We will update this citation with the additional citations mentioned in our Related Works.
>
> *making GP efficient (... for big data) ... considering today machine (not speaking about GPU using e.g. https://docs.jaxgaussianprocesses.com/) }*
>
> Regarding GPU, our code is fully implemented on GPU in Jax and exhibits large speed improvements compared to running on CPU. We leverage the `gpjax` package (an alternative to `jaxgaussianprocesses`) for the SVGP baseline, but decided against implementing our code using this package given that it was lacking some functionality for SSGPs/RFFs. Nonetheless, prior to this review period we did investigate implementation in `gpjax` and contacted to the maintainers for guidance in this regard, which we will consider for the future.
>
> *personally I would prefer to see the related work not in sec 4 but more at the beginning of the paper*
> Another reviewer echoed this comment, and we will move the section.
>
> *pseudo code would be nice within the paper*
> We can add pseudo-code either within the paper or in supplement, based on space, for a camera ready. If you have a strong preference that it should be in the main paper then we would appreciate that feedback.
>
> *please remove the unnecessary gray boxes in the plots*
> Apologies - we will make visualization edits and check that they are readable in black & white before a potential camera-ready.
>
> ## References
> Benton, Gregory W., Wesley J. Maddox, Jayson P. Salkey, Júlio Albinati, and Andrew Gordon Wilson. 2019. “Function-Space Distributions over Kernels.” In _Proceedings of the 33rd International Conference on Neural Information Processing Systems_. Curran Associates Inc.
>
> Hamid, Saad, Sebastian Schulze, Michael A. Osborne, and Stephen Roberts. 2022. “Marginalising over Stationary Kernels with Bayesian Quadrature.” In _Proceedings of The 25th International Conference on Artificial Intelligence and Statistics_. PMLR.

---

### Official Review · Reviewer_Y8NN · 2024-03-20

**Q2-1 Originality-Novelty:** 3
**Q2-2 Correctness-Technical Quality:** 3
**Q2-5 Clarity Of Writing:** 3

**Q1 Summary And Contributions:**

- Shows how the frequencies that make up the spectral approximation of the GP kernel can have a Bayesian treatment using the same Stein variational gradient descent.
- Leads to a posterior that includes the uncertainty in the kernel itself (this is the result of the M-SRFR part of the paper).

**Q2-3 Extent To Which Claims Are Supported By Evidence:**

2: Fair: the main claims are somewhat supported by evidence (but the experimental evaluation may be weak, or does not match entirely with the claims, important baselines may be missing, proofs contain important ideas but lack rigor, algorithmic details are only discussed superficially, references are imprecise, assumptions are not sufficiently motivated or explicated, etc.).

**Q2-4 Reproducibility:**

3: Good: key resources (e.g. proofs, code, data) are available and key details (e.g. proofs, experimental setup) are sufficiently well-described for competent researchers to confidently reproduce the main results.

**Q3 Main Strengths:**

- A novel and potentially quite useful approach to perform inference while also inferring a posterior over the kernel's spectrum.
- The mathematical steps appear correct.
- It is all very well and precisely written.
- The first few pages helped understanding.
- Some promising results.

**Q4 Main Weakness:**

- In terms of explanation: I feel like too many pages were spent on the preliminaries, and then the intuition one needed to understand the novel parts was very compact (I think from 3.3 onwards). I think people will struggle to follow some of it.
- Why was NLPD not used for assessing the quality of the posterior distribution (instead of calibration) & why did this do relatively poorly? Is there a sort of over-flexible-model problem causing an issue with the posterior being too broad? + What is the dimensionality of these datasets?
- Can you show the resulting kernel (distributions) for the example problems?

**Q5 Detailed Comments To The Authors:**

- The initial result (that one can use just the gradients of the kernel's spectral measure) feels lost in the middle of the paper. It would be really good if this could be highlighted, and some discussion around time complexity included? I don't see any experiments addressing just that aspect? But I might have completely misunderstood (hence why I've put this here and not as a main-weakness).
- Abstract: "Quality of the generated samples" seems a bit vague/confusing - not clear what it means.
- footnote on p2: Where is the '^1' in the text?
- in the contributions the text to the left of M-SRFR could have 'Regression' added to match the acronym?
- Why is Phi in (6) made of regular cos and sin? How does the Rahimi et al (2007) approach of using random bases cos(1/lambda w_m^x + b_m) so each one has a random phase. I'm sure there's a good reason for not using that here (I was just wondering/commenting).
- p4 "Equation 20" is mentioned a couple of times, and I think you might be referring to (18)? Also elsewhere you have used the (123) notation for referring to equations, rather than "Equation 123".
- Figure 1 is really useful, but could do with a longer caption. Why are the sampled kernels so varied (it feels there's enough data that they would be quite tight, also the periodic nature made me think that there might be a periodic aspect to the learned [distribution of] kernels?)
- what does the # symbol mean in Q_t := T_g_t # Q_{t-1} page 5 (section 3.3), just before (21).
- I would have found it quite helpful to have a bit more explanation around the start of Section 3.4 -- maybe some intuition around what the frequency matrix (Omega) means, for example.
- The RMSE improvements are reassuring, if quite surprising. What is the reason for this? What do the resulting kernels look like (is that something you can visualise?)
- Some terms are quite difficult for a non-expert to understand (maybe though, those more expert in this field will not have problems), e.g. "base distribution p_0" [below (13)] etc.
- It might also be good to demonstrate the equivalence with SVGD after (18) presumably the result is (17) but with the repulsive term added?

**Q9 Complying With Reviewing Instructions:**

Yes

---

> ### Author Rebuttal · Authors · 2024-04-05
>
> ## Clarity in Section 3
> - *... too many pages were spent on the preliminaries ... novel parts was very compact ...*
> - *... a bit more explanation around the start of Section 3.4*
> - *I think the explanation could be clearer in places*
>
> Thank you for the suggestions here - We aim for clarity and active discussion with reviewers on this matter.
>
> Choosing between theoretical underpinnings and practical applications was challenging, given our concern that many readers may not be familiar with functional kernel learning. This influenced our decision to provide a more extensive preliminary section than usual. We are open to suggestions for improving the theoretical and empirical connection and will refine this section for the camera-ready version to enhance clarity and flow.
>
> ## Further Highlighting of SVGD Kernel Approximation Sub-Result
> - *The initial result (that one can use just the gradients of the kernel's spectral measure) feels lost in the middle of the paper.*
> - *... good if this could be highlighted ... discussion around time complexity*
> - *I don't see any experiments addressing just that aspect?*
>
> The SVGD approximation to known kernels is a supportive result of our work, which is experimentally validated in Section 5.1, where SVGD outperforms baselines.
>
> We prioritized detailing our main contribution, functional kernel learning with M-SRFR, leading to the lesser emphasis on this secondary result. For the camera-ready version, we plan to include a concise explanation in the methods section and detailed analysis in the supplement, covering complexity and convergence, easily extendable from existing SVGD literature.
>
> ## Visualizing the M-SRFR Mixture
> - *Can you show the resulting kernel (distributions) for the example problems?*
> - *What do the resulting kernels look like (is that something you can visualise?)*
>
> Modifying a response to another reviewer here: We have produced an example of the learned kernels for the airfoil problem with $\alpha = 1.6$ at this [link](https://anonymous.4open.science/r/m-srfr/experiments/figures/airfoil_msrfr_kernel.pdf), which shows that despite equal initializations and a standard Gaussian prior, the kernels across mixtures begin to diversify in both mean and standard deviation. This behavior is also represented in Figure 1 in the paper, for which we will add a longer caption. Additionally, while these kernels may sometimes look similar to the eye, they can result in very different predictive distributions, for which we have included samples at this [link](https://anonymous.4open.science/r/m-srfr/experiments/figures/airfoil_preds.pdf).
>
> In general, we will observe differing behaviors (convergence to single mode / multi-modality) based on the problem and M-SRFR hyperparameter values, and we will look to include similar plots to that included here for all problems in the supplement to a camera-ready paper.
>
> ## Diving Deeper on Experimental Results
> - *The RMSE improvements are reassuring, if quite surprising. What is the reason for this?*
> - *[Figure 1] Why are the sampled kernels so varied ... there might be a periodic aspect to the learned [distribution of] kernels?*
>
> The RMSE improvements of our method are indeed quite reassuring, and we believe are a strong argument in favor of acceptance.
>
> In multi-modal scenarios, like the Mauna Loa problem with its multiple periodic trends, M-SRFR optimizes mixture components to focus on specific outcomes, enhancing the predictive distribution's breadth once combined.
>
> In general, our experiments showed single-kernel SSGPs are prone to local minima and initial conditions, contrasting with the mixture approach's robustness, leading to more accurate combined predictions. This behavior is demonstrated in the provided graph [here](https://anonymous.4open.science/r/m-srfr/experiments/figures/airfoil_preds.pdf) (a good example are the SSGP vs M-SRFR predictions for $y_{144}$ on the far right) which we'll duplicate for all experiments in the camera-ready supplement.
>
> ## Calibration Results
> - *Why was NLPD not used for assessing the quality of the posterior distribution (instead of calibration)*
> - *why did [calibration] do relatively poorly? Is there a sort of over-flexible-model problem causing an issue with the posterior being too broad?*
>
> Thank you for the suggestion - we will incorporate NLPD in a potential camera-ready. The calibration challenge for M-SRFR on some problems may stem from the choice of calibration metric. Given the multi-modal and diverse nature of the M-SRFR posterior, arising from the $M$ components in prediction, this issue is particularly evident in cases like the wine problem where components make precise yet narrow predictions, rendering Gaussian calibration unsuitable for the multi-modal mixture output.
>
> Adopting NLPD should offer a more appropriate evaluation metric, potentially enhancing M-SRFR's performance in uncertainty calibration.

---

### Official Review · Reviewer_toAg · 2024-03-21

**Q2-1 Originality-Novelty:** 3
**Q2-2 Correctness-Technical Quality:** 3
**Q2-5 Clarity Of Writing:** 2

**Q1 Summary And Contributions:**

The paper introduces a new technique called Stein random features (SRF) to enhance the scalability and flexibility of Gaussian processes (GPs) in large-scale regression tasks. By using Stein variational gradient descent, SRF generates high-quality random Fourier feature (RFF) samples with known spectral densities, improving sample accuracy and reliability. SRF efficiently approximates non-analytical spectral measure posteriors, offering more flexibility in modeling kernel properties. SRF's unique advantage lies in its use of log-probability gradients for both kernel approximation and Bayesian kernel learning, leading to superior performance compared to traditional methods in GP-related tasks. Empirical validation against baseline methods on kernel approximation and GP regression problems demonstrates SRF's effectiveness.

**Q2-3 Extent To Which Claims Are Supported By Evidence:**

3: Good: the main claims are supported by convincing evidence (in the form of adequate experimental evaluation, proofs, (pseudo-)code, references, assumptions).

**Q2-4 Reproducibility:**

3: Good: key resources (e.g. proofs, code, data) are available and key details (e.g. proofs, experimental setup) are sufficiently well-described for competent researchers to confidently reproduce the main results.

**Q3 Main Strengths:**

1. The paper presents SRF as a method for generating high-quality RFF samples with known spectral densities. By leveraging Stein variational gradient descent, SRF improves the sampling process, ensuring the accuracy and reliability of the generated samples.

2. SRF enables flexible and efficient approximation of traditionally non-analytical spectral measure posteriors. This allows for more accurate modeling and learning of the kernel properties, enhancing the performance of GPs in regression tasks.

3. SRF utilizes log-probability gradients for both kernel approximation and Bayesian kernel learning.

**Q4 Main Weakness:**

The paper lacks a discussion of the limitations and potential challenges associated with the proposed approach.

**Q5 Detailed Comments To The Authors:**

1. The numerical experiments lack specifics regarding the hyper-parameter selection.

2. Some details on the datasets should be provided, such as size, dimension, etc.

3. It would be beneficial to include additional remarks that explicitly illustrate how the conducted experiments demonstrate the effectiveness of the proposed approach, e.g., what's the meaning of calibration?

**Q9 Complying With Reviewing Instructions:**

Yes

---

> ### Author Rebuttal · Authors · 2024-04-04
>
> ## Stating the Limitations of M-SRFR
> - *The paper lacks a discussion of the limitations and potential challenges associated with the proposed approach.*
>
> While we believe that we begin to address these topics in Section 3.5 regarding the additional computational complexity induced by M-SRFR, as well as our analysis of experiments and shortcomings Section 5, we acknowledge your comment and will look to include further detail in a camera-ready. We will provide here a brief description of limitations with the intent to expand more formally in a camera-ready.
>
> Beyond the aforementioned complexity of training and performing inference with $M$ (mixture components) GPs, our choice of MMD distance between particle matrices $\Omega$ induces additional computation time. In practice, we find that using few mixture components $M < 10$ is sufficient for performance improvements in our experiments, but for large $M$ MMD approximation methods might serve as a recourse.
>
> As a general note, both SVGD and GP methods more broadly are known to degrade in higher dimensions, and our methods are not immune to this issue. As such, our method would likely require dimensionality reduction for high $d$ problems, but functions identically once reduction has been performed.
>
> Lastly, the RFF paradigm in general has advantages and disadvantages over other low-rank GP methodologies such as Nystroem approximation, but remains a valuable tool for many problems -- particularly for problems with a periodic nature such as time-series analysis and physical systems.
>
> ## Specifics on Experimental Details
> - *The numerical experiments lack specifics regarding the hyper-parameter selection.*
> - *Some details on the datasets should be provided, such as size, dimension, etc.*
>
> We will provide further details on these topics in the supplement for a camera-ready, and will describe them briefly here.
>
> Borrowing from another response, for M-SRFR, we performed hyperparameter optimization over $\alpha \in [0, 3]$ and $Q \in \{1, ..., 10\}$, as well as other parameters such as annealed-SVGD rate, learning rates, and training epochs, using 5-fold cross-validation. The hyperparameter tuning was rather minimal for our methods, with search being carried out through tree-based Parzen estimation with the `optuna` Python package with $< 100$ trials for all experiments. Even with few hyperparameter trials, we observe superior performance of M-SRFR. We performed similar hyper-parameter optimization for baseline models for their relevant parameters in order to provide a fair comparison between methods.
>
> The dataset dimensionality is as follows:
> - airfoil: $N = 1503, d = 6$
> - concrete: $N = 1030, d = 8$
> - energy: $N = 768, d = 16$
> - wine: $N = 1599, d = 11$
>
> We additionally are planning to add another large-scale experiment, potentially in spatiotemporal climate modeling, with $N$ on the scale of true large-scale regression problems.
>
> ## Experimental Metrics
> - *what's the meaning of calibration?*
>
> Calibration measures the proportion of true solutions that fall within 2 standard deviations of the predictive distribution generated by a GP. By properties of the predictive distribution, which is Gaussian, a well-calibrated model should have 95% calibration by this measure. Calibration serves to measure not only how well the mean prediction fairs against true data, but also how accurate the GP uncertainty estimation is.
>
> This is a crucial benefit of Bayesian methods, and as a result we report calibration results. We will be sure to include a discussion on this topic in a camera-ready. Additionally, based on a comment by another reviewer, we will add NLPD as a metric for uncertainty calibration.

---

### Official Review · Reviewer_m1ip · 2024-03-22

**Q2-1 Originality-Novelty:** 3
**Q2-2 Correctness-Technical Quality:** 3
**Q2-5 Clarity Of Writing:** 4

**Q1 Summary And Contributions:**

This paper presents a method based on Stein variational gradient descent for improving random Fourier features (RFF) in large-scale regression problems. Empirical results demonstrate very good results in kernel approximation and Bayesian kernel learning with this method.

**Q2-3 Extent To Which Claims Are Supported By Evidence:**

4: Excellent: all claims are supported by very convincing evidence (in the form of comprehensive experimental evaluation, rigorous mathematical proofs, detailed (pseudo-)code, precise references, well-motivated and realistic assumptions) and the authors deliver what they promise.

**Q2-4 Reproducibility:**

4: Excellent: key resources (e.g. proofs, code, data) are available and key details (e.g. proof sketches, experimental setup) are comprehensively described for competent researchers to confidently and easily reproduce the main results.

**Q3 Main Strengths:**

- This paper proposes a new GP method with random Fourier characteristics that are computed with Stein's variational gradient descent. The method seems very efficient and theoretically sound
- I like the idea of using SVGD in the space of distribution whose kernel is defined with a maximum discrepancy distance.

**Q4 Main Weakness:**

- I find quite unclear in Section 3.4 how the M different frequency matrices of the SVGD population are combined together to make the final prediction.
- a sensitivity analysis of the key parameters  alpha (equation 26),  governing the exploration/exploitation tradeoff in the method, is missing.
- a sensitivity analysis also of the parameter M, number of individuals in the population of distributions, would have also been interesting.

**Q5 Detailed Comments To The Authors:**

- when you use the M-SRFR methods, do you observe that SVGD converges towards different modes of the distribution of frequency matrices \Omega_m ? Or all the different frecuency matrices converge toward the same point ?

Minor remark :
In Sections 3.1 and 3.2, it seems that the reference to Equation 20 is incorrect, because this equation appears further down in the text.

**Q9 Complying With Reviewing Instructions:**

Yes

---

> ### Author Rebuttal · Authors · 2024-04-04
>
> ## Ablation Studies on M-SRFR Hyperparameters
> You included two comments relating to ablation/sensitivity analyses on parameters $M$ (number of mixture components) and $\alpha$ (repulsive force temperature) in our implementation. To select these values, we performed hyperparameter optimization over $\alpha \in [0, 3]$ and $Q \in \{1, ..., 10\}$, as well as other parameters such as annealed-SVGD rate, learning rates, and training epochs, using 5-fold cross-validation. The hyperparameter tuning was rather minimal for our methods, with search being carried out through tree-based Parzen estimation with the `optuna` Python package with $< 100$ trials for all experiments. We performed similar hyperparameter optimization for baseline models for their relevant parameters in order to provide a fair comparison between methods. In a camera-ready version, we can include full ablation studies on these values, but we can reasonably assume that in practice these values will be selected through hyperparameter optimization as we have done here.
>
> ## Producing M-SRFR Predictions
> - *I find quite unclear in Section 3.4 how the M different frequency matrices of the SVGD population are combined together to make the final prediction.*
>
> You are correct to point out that our explanation in our submission is rather limited, and we will include further details in a camera-ready version. In summary for the M-SRFR model, we train $M$ GPs using the M-SRFR update rule in Definition 3.2. This results in $M$ GPs with differing kernels (defined by their RFF frequencies $\mathbf{\Omega}_m$). For each GP, we calculate the GP predictive distribution $p_m(y^* | x^*, D, \mathbf{\Omega}_m) \sim \mathcal{N}(y^* | \mu_m, \sigma_m)$, where $\mu_m$ and $\sigma_m$ are calculated as in traditional GPs.
>
> We then generate $L$ (in our experiments we use $L = 100$) samples from each $p_m$, resulting in $L \times M$ total samples $\hat{y}^*_{lm}$. The final prediction which combines the mixture components is then approximated as $p(y^* |x^*, D, \mathbf{\Omega}) \sim \mathcal{N}(y^* | \mu(\hat{y}^*_{lm}), \sigma(\hat{y}^*_{lm}))$, ie. the empirical mean and standard-deviation of the $L \times M$ samples. We follow this procedure in order to capture the potentially multi-modal nature of the predictive distribution implied by using $M$ mixture components. This is the same approach followed by Pinder et al. [2022], who apply SVGD for variational approximation to a GP posterior.
>
> We can alternatively view this procedure as producing predictions from an $M$-component Gaussian mixture distribution with equal weights between components. We will formally include the details described here in a camera-ready paper.
>
> ## Visualizing the M-SRFR Ensemble
> - *do you observe that SVGD converges towards different modes of the distribution of frequency matrices $\Omega_m$? Or all the different frequency matrices converge toward the same point?*
>
> The answer to this question is difficult to categorically answer as it will depend on several components, including the initialization method of each mixture component kernel, repulsive temperature hyperparameter $\alpha$, the selected prior on $p(\omega)$ and the underlying multi-modality of the true data $\mathbf{y}$. Theoretically, we can tailor the initialization of $\mathbf{\Omega}_m$ and repulsive temperature $\alpha$ to encourage or discourage exploration.
>
> In our experiments, we optimize $\alpha$ as a hyperparameter and initialize all $M$ mixtures as standard RBFs, and use a Gaussian prior for $p(\omega)$. We have produced an example of the learned kernels for the airfoil problem with $\alpha = 1.6$ at this [link](https://anonymous.4open.science/r/m-srfr/experiments/figures/airfoil_msrfr_kernel.pdf), which shows that despite equal initializations and a standard Gaussian prior, the kernels across mixtures begin to diversify in both mean and standard deviation. This behavior is also represented in Figure 1 in the paper. Additionally, while these kernels may sometimes look similar to the eye, they can result in very different predictive distributions, for which we have included samples at this [link](https://anonymous.4open.science/r/m-srfr/experiments/figures/airfoil_preds.pdf).
>
> In general, we will observe differing behaviors (convergence to single mode / multi-modality) depending on all the previously mentioned factors, and we will look to include similar plots to that included here for all problems in the supplement to a camera-ready paper.
>
> ## Misc
> *Minor remark : In Sections 3.1 and 3.2, it seems that the reference to Equation 20 is incorrect, because this equation appears further down in the text.*
> Thank you for noting this - this was indeed a typo which we have now fixed.
>
> ## References
> Pinder, Thomas, Christopher Nemeth, and David Leslie. 2020. “Stein Variational Gaussian Processes.” [https://arxiv.org/abs/2009.12141v3](https://arxiv.org/abs/2009.12141v3).

---

### Official Review · Reviewer_Y1Tf · 2024-03-22

**Q2-1 Originality-Novelty:** 3
**Q2-2 Correctness-Technical Quality:** 3
**Q2-5 Clarity Of Writing:** 3

**Q1 Summary And Contributions:**

This paper applies Stein variational gradient descent (Liu and Wang, 2016) to the problem of kernel approximation and Gaussian process regression with a learned kernel.

The main ideas:

(1) They show that SVGD can be used to efficiently learn a high-quality particle approximation of the kernel measure, by analytically characterizing the gradient (Thm. 3.1).

This allows an efficient approximation of the kernel matrix with a small number of particles, which can be used to speed up inference. The approximation is argued to be higher quality than existing particle approximations of the kernel measure.

(2) By choosing an appropriate prior over the set of measures, SVGD also allows us to do posterior inference on the kernel--this gives us a GP regression with a "learned kernel"

They illustrate how may produce a better low-rank approximation to the kernel, and how it can be used to improve upon GP regression.

**Q2-3 Extent To Which Claims Are Supported By Evidence:**

3: Good: the main claims are supported by convincing evidence (in the form of adequate experimental evaluation, proofs, (pseudo-)code, references, assumptions).

**Q2-4 Reproducibility:**

4: Excellent: key resources (e.g. proofs, code, data) are available and key details (e.g. proof sketches, experimental setup) are comprehensively described for competent researchers to confidently and easily reproduce the main results.

**Q3 Main Strengths:**

The ideas presented in the paper are novel and useful.

They give a practical method for doing inference on the covariance kernel, which outperforms competing approaches.

They also show how SVGD can be used to approximate of the kernel measure, which is interesting by itself.

**Q4 Main Weakness:**

- One of the main claims of the paper is that using SVGD should improve the particle approximation to the kernel measure. However, in the experiments presented, the SVGD-SSGP algorithm does not seem to perform better than any of the competing methods. It would be good to at least discuss this point.

- It would also be helpful to include the full spectrum GP for comparison, and to discuss / compare to other popular ways of approximating the kernel Gram matrix (row/column sketching, Nystrom approximation). These methods should probably be more extensively discussed (e.g., Yang Pilanci Wainwright '17 is not cited).

- Based upon Figure 2, it seems that the SVGD particle approximation is not as good as the Nystrom approximation, which is a well understood technique.

**Q5 Detailed Comments To The Authors:**

The organization of the paper is a bit confusing. I would move "Related Work" up to be near the beginning, because it currently disrupts the flow of the paper.

In Section 3.5 (pg. 6), you use "Q" to refer to the number of mixture components, though "Q" was previously used to refer to the mixing distribution, and "M" was the number of components?

**Q9 Complying With Reviewing Instructions:**

Yes

---

> ### Author Rebuttal · Authors · 2024-04-04
>
> ## Kernel Approximation Results and Baselines
> We will address the following comments here.
> - *The SVGD approximation to the kernel measure, by itself, is less interesting since it doesn't seem to improve upon the Nystrom approximation.*
> - *Based upon Figure 2, it seems that the SVGD particle approximation is not as good as the Nystrom approximation, which is a well understood technique.*
>
> We note that while it is true that in Figure 2 SVGD underperforms Nystroem when $\frac{R}{N} < 0.2$, above this threshold it outperforms Nystroem. SVGD is known to scale poorly with dimensionality, and in high dimensions regular MC (not QMC) tends to admit the best RFF approximation error. However, as we note in the paper, there is a general degeneracy of kernel-based methods in high-dimensions. As a result, we would anticipate that a form of dimensionality reduction is applied in pre-processing for true high-dimensional kernel regression/GP methods.
>
> In a camera-ready, we will additionally look to recreate the study represented in Figure 2 over $N$, as for data-based methods such as Nystroem, as we expect data-based kernel approximation methods, which are inherently local, to exhibit worse scaling than RFF methodologies, which operate globally due to their formulation in frequency space, as $N$ grows.
>
> ## Clarifying The SVGD-SSGP Baseline vs Our Method (M-SRFR)
> - *One of the main claims of the paper is that using SVGD should improve the particle approximation to the kernel measure. However, in the experiments presented, the SVGD-SSGP algorithm does not seem to perform better than any of the competing methods. It would be good to at least discuss this point.*
>
> We would like to draw the distinction between the tasks of kernel approximation and kernel learning. The sub-result in Section 5.1, and discussed above, shows SVGD's strong performance for kernel approximation of known kernels. For kernel learning, however, we focus on the M-SRFR method and the strong performance on regression problems.
>
> For kernel learning, we intentionally do not focus our contribution on SVGD-SSGP, which we include in experiments primarily as a baseline to demonstrate that the M-SRFR method has advantages over point-estimate functional kernel learning with SVGD. We will make this more clear in a potential camera-ready paper.
>
> For the purpose of discussion and potential future work in functional kernel learning, our hypothesis as to why SVGD-SSGP underperforms compared to a traditional RFFs primarily stems from the geometry of spectral distributions. SVGD-SVGP is essentially a traditional SSGP with entropy regularization (through the SVGD kernel repulsion term) and Hilbert space smoothing (through the attractive term). Optimal kernels for real data are often sharply peaked in the frequency domain, with minor variations resulting in significantly different kernels in the spatial/time domain. As a result, we suspect both the smoothing and repulsive terms discourage the frequencies away from their optimal values in the kernel learning setting. This would explain why SVGD does well at approximating the RBF, which has a smooth spectral distribution, but SVGD-SSGP underperforms in the kernel learning setting.
>
> We note however that this issue is not present within the M-SRFR setup, where the smoothing and repulsion terms operate between matrices of particles. This is a weaker force on the individual frequencies within each matrix, and serves more to encourage diversity between mixture kernels. This would help explain, along with the benefits inherent to the mixture model approach, why we do not observe degeneracy of M-SRFR in the regression setting.
>
> ## Alternative Baselines
> - *It would also be helpful to include the full spectrum GP for comparison*
> - *... compare to other popular ways of approximating the kernel Gram matrix (row/column sketching, Nystrom approximation)*
>
> Our main result (M-SRFR) is focused on the joint low-rank kernel-approximation *and* learning setting inherent to the SSGP formulation. That being said, we include a Nystroem-based GP in regression experiments in the stochastic variational GP (SVGP).
> We think that SSGPs offer unique benefit in both their capacity to perform kernel learning and perform well on problems with periodic structure, but based on yours and other reviewers' comments, we will look to either further discuss or add as baselines alternative low-rank methods in a camera-ready. We can certainly add a full-spectrum GP for comparative purposes.
>
> ## Misc
> *I would move "Related Work" up to be near the beginning*
> Thank you for the suggestion - we will move the Related Work to after the Introduction.
>
> *you use "Q" to refer to the number of mixture components, though "Q" was previously used to refer to the mixing distribution, and "M" was the number of components?*
> Thank you for pointing this out - this is a typo and a result of a previous notation choice. We have implemented your suggestion and harmonized notation.

---

### Official Review · Reviewer_s7Xr · 2024-04-01

**Q2-1 Originality-Novelty:** 3
**Q2-2 Correctness-Technical Quality:** 3
**Q2-5 Clarity Of Writing:** 3

**Q1 Summary And Contributions:**

The paper proposes an approach that utilizes Stein variational gradient descent (SVGD) on the space of measures to learn the Kernel for random features regression in a Bayesian manner. SVGD is additionally used to obtain the random features approximation by sampling from the corresponding spectral density. The Kernel is learned by lifting the optimization over Kernel to the space of spectral measures. SVGD over matrix valued samples, representing empirical measures is then utilized to sample from the posterior over spectral measures. The paper includes promising results supporting the effectiveness of the approach.

**Q2-3 Extent To Which Claims Are Supported By Evidence:**

3: Good: the main claims are supported by convincing evidence (in the form of adequate experimental evaluation, proofs, (pseudo-)code, references, assumptions).

**Q2-4 Reproducibility:**

3: Good: key resources (e.g. proofs, code, data) are available and key details (e.g. proofs, experimental setup) are sufficiently well-described for competent researchers to confidently reproduce the main results.

**Q3 Main Strengths:**

- The proposed approach is novel and well-motivated.
- The paper provides an adequate discussion of the background and motivation for the proposed approach.
- The approach is supported by experimental comparisons against baselines.

**Q4 Main Weakness:**

- The connections between Kernel learning and SVGD need to be further justified since the kernel in SVGD is fixed and doesn't depend on the sampling measure while the Kernel learning update in eq. 17 involves a Kernel dependent on the spectral measure $\pi$.
- The paper lacks theoretical guarantees or an emprical analysis of convergence and the number of points in the empirical measures and the number of draws of the empirical measures required to acheive good performance.
- Sections 3.1-3.4 are hard to follow, especially since Definition 3.2 is not supported by an analysis or a sketch.
- The use of SVGD for sampling from the spectral measure is not particularly novel since SVGD is a well-known sampling algorithm. Therefore, the major contribution lies in the use of SVGD for the learning of the Kernel.

**Q5 Detailed Comments To The Authors:**

- In eq 17, doesn't the kernel depend on $\pi$ ? This makes it different from SVGD.
- In eq. 17, the $\approx$ is precisely in which sense?
- How does Definition 3.2 follow from Theorem 3.1?
- How is the Kernel in eq. 28 chosen?

**Q9 Complying With Reviewing Instructions:**

Yes

---

> ### Author Rebuttal · Authors · 2024-04-04
>
> We thank you for providing detailed comments and critiques. We have addressed the main themes of your review here.
>
> ## Clarification on GP Kernel $k$ vs. SVGD Kernel $\kappa$ (kappa).
> The following two comments represent a potential misunderstanding which we would like the clarify:
> - *The connections between Kernel learning and SVGD need to be further justified since the kernel in SVGD is fixed and doesn't depend on the sampling measure while the Kernel learning update in eq. 17 involves a Kernel dependent on the spectral measure.*
> - *In eq 17, doesn't the kernel depend on $\pi$? This makes it different from SVGD.*
>
> In our methods, there are two different kernels: the GP kernel $k$, subject to approximation or learning in our methods, and the SVGD kernel $\kappa$ (kappa). Unlike $k$, which is learned, $\kappa$ is fixed and utilized within SVGD to manage particle interactions, and does **not** depend on RFF frequencies $\omega$. Therefore, Eq 17, when enhanced by a repulsive term in Eq 18, demonstrates standard SVGD application. We apologize for any confusion, as there is a typo in the paragraph following Eq 17 in which we use $k$ instead of of $\kappa$, which we have now fixed.
>
> Section 3.1 highlights an equivalence between functional kernel learning and SVGD application to RFF frequencies $\omega$, with the main difference from vanilla SVGD lying in the modified SVGD target, which we modify to be a nonlinear functional of the GP-NLL w.r.t kernel spectral measure  $\pi(\omega)$. This approach has been well-motivated by previous works (Wang and Liu, 2019, Mallick et al, 2021, in main text References) on *nonlinear* SVGD.
>
> To avoid confusion, we acknowledge the potential overlap in notations for $k$ and $\kappa$ and will consider revising them in the final version.
> ## Connecting Theorem 3.1 to Definition 3.2
> With the aforementioned clarification on $k$ vs. $\kappa$ (kappa) in order, we wish to address the following comments and provide more information:
> - *Definition 3.2 is not supported by an analysis or a sketch.*
> - *... lacks details about the connections between the main theorem and the proposed algorithm.*
> - *How does Definition 3.2 follow from Theorem 3.1?*
>
> The goal of Sections 3.2-3.4 is to move beyond the idea of learning a single GP kernel defined by $\pi(\omega)$ with SVGD to instead learn a "posterior" $P(\pi | D)$ over the space of probability measures $\pi$.
>
> This is not in general a tractable problem, given that there is no intuitive way in which we can define a parametric form for a distribution over all probability measures. Thus, Thm 3.1 presents an iterative functional-SVGD scheme to perturb a set of initial measure samples $\pi_m$ in the direction defined by Eqn 24 to asymptotically arrive at an empirical approximation to $P(\pi | D)$. Each "measure particle" in this SVGD scheme now represents a given probability measure $\pi_m$.
>
> Section 3.4 and Definition 3.2 provide the final step to make this inference tractable by representing each "measure particle" empirically by the matrix $\mathbf{\Omega}_m$, where each row of $\mathbf{\Omega}_m$ corresponds to a sample of $\pi_m(\omega)$. The scheme of Def 3.2 can be used to learn $M$ GP kernels that, like in traditional SVGD, concentrate on areas of high posterior likelihood while encouraging diversity between the $M$ kernels. We again note that the kernel $\kappa$ (kappa) used in this scheme is separate from the $M$ learned GP kernels, and does not depend on frequencies $\omega$.
>
> We understand how this framing can induce confusion in readers given the novelty of the nonlinear-SVGD method. We went back and forth during writing on the degree we focused on the theoretical underpinnings or on practical application, and have quite a bit of content not included in this draft that we can exchange to help strike a better balance given feedback. To this end, we welcome your suggestions as to how to improve upon the flow and connection between the theoretical and empirical given the above description. Regardless, we hope that the connection between Theorem 3.1 and Definition 3.2 has been made more clear to your as a result of this response.
>
> ## Convergence and Error Analysis
> - *The paper lacks theoretical guarantees or an empirical analysis of convergence...*
> - *... without precise mathematical statements of equivalence*
>
> While we believe that our current contribution which justifies the theoretical underpinnings of M-SRFR is enough to merit acceptance, we will also seek to include convergence (theoretical and empirical) analysis for our method in a camera-ready. We hope that our current theoretical justifications in the main body and supplement are indicative of our ability to produce these further results.
>
> ## Misc
> *In eq. 17, the $\approx$ is precisely in which sense?*
> The approximation comes from the fact that measure $\pi$ is variationally approximated as $q$. In the SVGD case, this is an empirical measure formed by a set of $R$ particles.

---

### Meta-Review · Area_Chair_a9kZ · 2024-04-15

The paper proposes an approach that combines Stein variational gradient descent (SVGD) with kernel learning for random features regression, offering potential advancements in the Bayesian methodology for kernel approximation. The claims are found by the referee to be supported by convincing theoretical insights and empirical evaluations.

The authors effectively use SVGD to learn the kernel by optimizing over the space of spectral measures, which has been noted as a strength of the paper. The inclusion of both theoretical developments and experimental validations enriches the paper's contribution to the field. Furthermore, the reproducibility of the results is well-supported by the availability of key resources such as proofs, code, and data.

Despite some minor weaknesses in the connection between kernel learning and SVGD, and the lack of detailed theoretical guarantees for convergence, the consensus among the reviewers is that the paper's strengths outweigh its limitations. The authors have also responded adequately in their rebuttal, clarifying misunderstandings and agreeing to refine the presentation .

Given the originality of the approach, the soundness of the technical content, and the potential impact on future research in kernel methods and Bayesian regression, the paper has been judged favorably by the referees.